# To Think or Not To Think, That is The Question for LLM Reasoning in Theory of Mind Tasks

## Abstract

Theory of Mind (ToM) assesses whether models can infer hidden mental states such as beliefs, desires, and intentions, which is essential for natural social interaction. Although recent progress in Large Reasoning Models (LRMs) has boosted step-by-step inference in mathematics and coding, it is still underexplored whether this benefit transfers to socio-cognitive skills. We present a systematic study of 11 advanced Large Language Models (LLMs), comparing reasoning models with non-reasoning models on three representative ToM benchmarks. The results show that reasoning models do not consistently outperform base models and sometimes perform worse. A fine-grained analysis reveals three insights. First, slow thinking collapses: accuracy significantly drops as responses grow longer, and larger reasoning budgets hurt performance. Second, moderate and adaptive reasoning benefits performance: constraining reasoning length mitigates failure, while distinct success patterns demonstrate the necessity of dynamic adaptation. Third, option matching shortcut: when multiple choice options are removed, reasoning models improve markedly, indicating reliance on option matching rather than genuine deduction. These results highlight the advancement of LRMs in formal reasoning (e.g., math, code) cannot be transferred to ToM, a typical task in social reasoning. We conclude that achieving robust ToM requires developing unique capabilities beyond existing reasoning methods and we provide a preliminary exploration of such an approach with a combination of Slow-to-Fast (S2F) adaptive reasoning and Think-to-Match (T2M) shortcut prevention.

## 1 Introduction

Theory of Mind (ToM) refers to the human capacity to infer the unobservable mental states of others, such as beliefs, desires, and emotions, forming the foundation of social cognition (Chen et al., 2025; Sarıtaş et al., 2025; Nguyen et al., 2025). It enables individuals to interpret subtle cues, anticipate behavior, and maintain meaningful communication. Research in large-scale reasoning has recently gained significant momentum (Guo et al., 2025; Yang et al., 2025). Breakthroughs in Large Reasoning Models (LRMs) have shown that enhancing step-by-step inference capabilities yields dramatic improvements in structured domains like mathematics, code generation, and scientific problem-solving. These advances suggest that explicit reasoning serves as a general catalyst, potentially elevating models beyond surface-level pattern recognition toward more systematic and reliable intelligence. Against this backdrop, a critical question arises: Can these powerful reasoning mechanisms, proven effective in analytical domains, be successfully transferred to enhance the socio-cognitive capabilities required for ToM in LLMs?

Existing research has only provided a preliminary exploration of the role of reasoning in ToM. The HiToM study demonstrated that applying Chain-of-Thought (CoT) prompting yields insignificant performance increases and may even amplify the model's susceptibility to deceptive information during testing (Wu et al., 2023). Subsequent research has focused on incorporating more structured, human-like reasoning processes. Strategies such as integrating perspective-taking and reflection into CoT templates have effectively enhanced performance on ToM benchmarks (Zhou et al., 2023; Wilf et al., 2024; Wang & Zhao, 2024). More recently, significant attention has shifted towards Reinforcement Learning (RL) methods like GRPO, introduced by DeepSeek-R1, prompting exploration into R1-style inherent reasoning capabilities (Guo et al., 2025; Wang et al., 2025; Zheng et al., 2025). However, the efficacy of this approach remains contested; the ToM-RL study found that small models

trained with GRPO can suffer from reasoning collapse, performing worse than those trained with standard Supervised Fine-Tuning (SFT) (Lu et al., 2025). The current focus on method-level enhancements sidesteps a more fundamental question: *To what extent does reasoning itself intrinsically contribute to ToM capabilities?* A systematic analysis contrasting the ToM performance of reasoning models against non-reasoning models is essential to isolate and understand the genetic impact of reasoning mechanisms.

To fill this gap, we conduct a comprehensive study on the effectiveness of LRMs on ToM tasks. Specifically, we evaluate model performance across three representative benchmarks, i.e., HiToM, ToMATO, and ToMBench, covering a range of reasoning order, taxonomy, and scenarios. We find that reasoning models (e.g., DeepSeek-R1, Qwen3-8B-Reasoning) generally fail to outperform their non-reasoning counterparts (e.g., DeepSeek-V3, Qwen3-8B). For example, Qwen3-8B-Reasoning achieves a score of 0.6478, which is significantly lower than the 0.7047 scored by its non-reasoning version on the latest ToMATO benchmark. This counterintuitive performance suggests that the inherent reasoning capability in reasoning models may not be effective for ToM tasks. To move beyond this observation, we provide a deeper analysis to diagnose when these failures occur and the underlying reasons why.

Our analysis reveals three fundamental insights behind models' reasoning errors: **(i) Slow thinking correlates with reasoning collapse.** We find that errors are heavily concentrated in longer responses, meaning the longer a model thinks, the more likely it is to fail. Pushing models like GPT-o3 and GPT-o4-mini to expend more reasoning effort actually backfires, leading to decreased performance. This demonstrates that for complex ToM tasks, prolonged computation is a liability, not an asset. **(ii) Moderate and adaptive reasoning benefits performance.** We find that applying CoT to non-reasoning models and limiting the reasoning length of reasoning models both lead to performance gains. Furthermore, our experiments reveal the complementary strengths of reasoning and non-reasoning models. Consequently, we identify moderate and adaptive reasoning as a promising direction for future ToM research. **(iii) Reasoning takes option matching rather than step-by-step deduction.** Our experiments on HiToM demonstrate that when we remove the multiple-choice options, the performance of reasoning models such as DeepSeek-R1 and Qwen3-8B-Reasoning dramatically improves. A look "under the hood" at their thinking process confirms why: they are not reasoning to a solution from the ground up, but rather matching the most likely answer from the provided list. This suggests their success often relies on option matching rather than genuine, step-by-step deduction.

The identified failure mechanisms of reasoning collapse and option matching shortcuts highlight the advancement of LRMs in formal reasoning (e.g., math, code) cannot effectively lead to increasing performance in ToM, a typical task in social reasoning. Our findings suggest that strategies beneficial in formal domains, such as prolonged deliberation, are often counterproductive in the ambiguous context of ToM. This implies that improving ToM is not about simply scaling existing analytical methods, but requires developing unique capabilities tailored for social reasoning. We leverage a combination of **S**low-to-**F**ast reasoning (**S2F**) and **T**hink-to-**M**atch (**T2M**) to provides a preliminary exploration into these necessary new approaches. Our contributions can be summarized as:

- We provide a systematic comparison of reasoning and non-reasoning models on ToM tasks, revealing the counterintuitive finding that reasoning models fail to establish advantages.

- We identify and provide empirical evidence for two core failure reasons in ToM reasoning: *slow thinking collapse*, where prolonged deliberation becomes counterproductive, and *option matching shortcut*, where models favor superficial pattern matching over genuine deduction.

- Based on these findings, we discuss the fundamental divergence between ToM and formal reasoning and provide a preliminary exploration of solutions, including the Slow-to-Fast adaptive reasoning method and Think-to-Match shortcut prevention.

## 2 RELATED WORK

**Theory of Mind Capability Evaluation.** The benchmarks in ToM are mainly based on the Sally-Anne test, following a multi-choice format (Sarıtaş et al., 2025; Nguyen et al., 2025). ToMi extends ToM-bAbi via procedurally varied narratives and a systematic sweep over reality, memory, and first-/second-order belief queries, while HI-TOM pushes the envelope to fourth-order belief reasoning

(Wu et al., 2023). Moving beyond templated narratives, FANTOM introduces dialogue-mediated settings and explicitly targets "illusory ToM," where responses appear correct yet violate underlying logical constraints (Kim et al.). In parallel, BigToM (Gandhi et al., 2023) and OpenToM (Xu et al., 2024) broaden the mental-state taxonomy to include percepts, desires, and emotions. Domain-specific evaluations have also emerged: NegotiationToM integrates belief–desire–intention (BDI) reasoning within multi-round bargaining dialogues (Chan et al., 2024), and ToMBench pursues near-comprehensive ATOMS coverage with bilingual construction to mitigate pretraining contamination (Chen et al., 2024). Complementary efforts explore search- and generation-centric data creation, including A*-driven diversification in ExploreToM (Sclar et al.) and LLM–LLM self-play with information asymmetry in ToMATO (Shinoda et al., 2025). To comprehensively assess reasoning effectiveness, we evaluate on HiToM, ToMBench, and ToMATO, which together span higher-order belief depth, a broad mental-state taxonomy, and diverse evaluation scenarios.

**Large Reasoning Model Evaluation.**   Benchmarks for LRMs span mathematics, formal logic, commonsense, code, and agentic interaction. Math suites range from contest-style and grade-school word-problem sets (MATH, GSM8K) to visually grounded mathematics (MathVista) and chart reasoning (ChartQA) (Hendrycks et al.; Cobbe et al., 2021; Lu et al.; Masry et al., 2022). Logical-reasoning datasets cover deductive and abductive regimes (ProofWriter, FOLIO) and relational induction stress tests (CLUTRR) (Tafjord et al., 2021; Han et al., 2024; Sinha et al., 2019). Commonsense resources probe physical plausibility and broad knowledge (WinoGrande, MMLU) (Sakaguchi et al., 2021; Hendrycks et al., 2021). Code benchmarks emphasize exactness and executability, from function synthesis to real-world issue resolution (HumanEval, MBPP, SWE-bench) (Chen et al., 2021; Austin et al., 2021; Jimenez et al., 2024). Finally, web/embodied environments evaluate multi-step planning and tool use under interaction (Mind2Web, ALFWorld) (Deng et al., 2023; Shridhar et al., 2021). We systematically evaluate LRMs on ToM tasks, diagnose the failure modes underlying their reasoning errors, and outline directions for strengthening social reasoning.

## 3   EXPERIMENTS

### 3.1   EXPERIMENTAL SETUP

To comprehensively study the effectiveness of reasoning, we select 11 models including (i) **Reasoning Models:** Claude-Sonnet-4, Grok-3-mini, GPT-o4-mini, GPT-o3, DeepSeek-R1, Qwen3-8B-Reasoning, and Qwen3-32B-Reasoning; (ii) **Non-Reasoning Models:** GPT-4o, DeepSeek-V3, Qwen3-8B, and Qwen3-32B. Our selection spans both matched reasoning and non-reasoning variants within the same families, as well as additional reasoning-only models. This design allows for analysis from both macro and micro perspectives. From a macro perspective, we compare the overall reasoning and non-reasoning groups; from a micro perspective, we investigate the specific differences between individual models. Unless not adjustable, all models use the same settings: temperature 0, top-p 1, and a maximum output length of 2048 tokens.

To systematically evaluate ToM, we select three specialized benchmarks, each designed to probe a distinct aspect of social reasoning. This ensures a multi-faceted assessment of model capabilities. (i) **HiToM** (Wu et al., 2023) focuses on the depth of reasoning. It tests a model's ability to handle complex, multi-level recursive beliefs (from 0th to 4th-order) in narratives that include deceptive agents. (ii) **ToMATO** (Shinoda et al., 2025) assesses ToM in realistic, interactive contexts. It uses conversation-based scenarios between role-playing agents to test how well a model can infer mental states from dynamic, ongoing dialogues. (iii) **ToMBench** (Chen et al., 2024) provides broad taxonomic coverage. It systematically evaluates a wide range of distinct mental states, including beliefs, desires, emotions, and intentions, ensuring a comprehensive assessment of ToM abilities. All the experimental results are evaluated by the accuracy. Full results are available in Appendix.

### 3.2   EXPERIMENTAL RESULTS

To intuitively compare reasoning and non-reasoning models, we present a side-by-side performance comparison of models from the same series. This analysis investigates whether a focus on reasoning leads to universal improvements in ToM tasks. Table 1 reveals a counterintuitive pattern: reasoning models fail to consistently outperform their non-reasoning counterparts. This trend is consistent across

Table 1: Overall results of all reasoning and non-reasoning models on three benchmarks

| Dataset | GPT | | | DeepSeek | | Qwen3-8B | | Qwen3-32B | |
|---|---|---|---|---|---|---|---|---|---|
| | GPT-4o | GPT-o4-mini | GPT-o3 | DeepSeek-V3 | DeepSeek-R1 | Qwen3-8B | Qwen3-8B-Reasoning | Qwen3-32B | Qwen3-32B-Reasoning |
| HiToM | 0.607 | 0.547 | **0.747** | **0.694** | 0.549 | **0.558** | 0.481 | 0.586 | **0.680** |
| ToMATO | **0.822** | 0.792 | 0.817 | **0.782** | 0.749 | **0.705** | 0.648 | **0.732** | 0.714 |
| ToMBench | 0.797 | 0.803 | **0.818** | 0.763 | **0.801** | 0.674 | **0.729** | 0.754 | **0.775** |

model families. While reasoning models retain a clear advantage on ToMBench, this superiority is completely reversed on ToMATO, where non-reasoning models achieve higher scores in every pairing. Similarly, HiToM presents mixed results, with non-reasoning models winning the majority of comparisons. These findings highlight that reasoning models do not guarantee success and can incur performance costs in specific contexts, prompting further questions on why reasoning fails. Detailed results are reported in Appendix B.

## 4 ANALYSIS & EXPLORATION

### 4.1 WHEN REASONING FAILS TO OUTPERFORM?

Table 2: Performance of models on HiToM (reasoning orders) and ToMBench (taxonomy categories).

| Model | HiToM | | | | | ToMBench | | | | | |
|---|---|---|---|---|---|---|---|---|---|---|---|
| | Order 0 | Order 1 | Order 2 | Order 3 | Order 4 | Belief | Desire | Emotion | Intention | Knowledge | Non |
| *GPT* | | | | | | | | | | | |
| GPT-o4-mini | **1.000** | 0.731 | 0.460 | 0.293 | 0.249 | 0.916 | 0.678 | 0.769 | 0.824 | **0.648** | 0.770 |
| GPT-o3 | 0.996 | **0.912** | **0.733** | **0.625** | **0.467** | **0.923** | **0.689** | **0.786** | **0.856** | 0.617 | **0.803** |
| GPT-4o | 0.979 | 0.692 | 0.571 | 0.408 | 0.383 | 0.909 | 0.661 | 0.762 | 0.847 | 0.624 | 0.782 |
| *DeepSeek* | | | | | | | | | | | |
| DeepSeek-R1 | **0.988** | **0.762** | 0.508 | 0.292 | 0.196 | **0.902** | 0.661 | **0.764** | **0.853** | **0.593** | 0.794 |
| DeepSeek-V3 | 0.979 | 0.650 | **0.600** | **0.633** | **0.608** | 0.812 | **0.683** | 0.738 | 0.838 | 0.479 | **0.814** |
| *Qwen3-8B* | | | | | | | | | | | |
| Qwen3-8B-Reasoning | **0.850** | **0.679** | 0.421 | 0.246 | 0.208 | **0.833** | **0.628** | **0.702** | **0.726** | 0.479 | **0.745** |
| Qwen3-8B | 0.846 | 0.667 | **0.529** | **0.379** | **0.367** | 0.656 | 0.611 | 0.674 | 0.715 | **0.555** | 0.737 |
| *Qwen3-32B* | | | | | | | | | | | |
| Qwen3-32B-Reasoning | **1.000** | **0.775** | **0.600** | **0.579** | **0.446** | **0.872** | 0.650 | **0.743** | 0.818 | **0.534** | **0.782** |
| Qwen3-32B | 0.971 | 0.629 | 0.546 | 0.379 | 0.404 | 0.846 | **0.689** | 0.695 | **0.824** | 0.462 | 0.775 |
| *Others* | | | | | | | | | | | |
| Claude-Sonnet-4 | 1.000 | 0.846 | 0.775 | 0.767 | 0.721 | 0.939 | 0.672 | 0.762 | 0.885 | 0.666 | 0.822 |
| Grok-3-mini | 0.925 | 0.319 | 0.197 | 0.188 | 0.173 | 0.889 | 0.622 | 0.745 | 0.850 | 0.493 | 0.837 |

Moving beyond overall performance comparison, this section provides a fine-grained analysis along two axes: the reasoning order and taxonomy. The benchmarks test different reasoning complexities, as ToMBench requires 1st-order reasoning, ToMATO up to 2nd-order, and HiToM up to 4th-order. Their taxonomic scope also expands from the belief-focused HiToM to the more comprehensive ToMBench, which covers 6 mental state dimensions. Our results confirm that model proficiency is not uniform: it degrades with higher-order reasoning and varies across different mental states.

> 💡 **Takeaway 1:** Reasoning models fail to demonstrate a definitive advantage, and different reasoning models exhibit divergent patterns in both overall and break-down accuracy scores in different reasoning orders and task types.

### 4.1.1 REASONING LOSES DOMINANCE IN HIGH-ORDER INFERENCE.

Table 2 shows the detailed performance on different reasoning orders. In low-complexity scenarios (Orders 0-1), reasoning models exhibit a significant performance advantage. For example, GPT-o3 maintains an accuracy above 0.9 on 1st-order tasks. However, this trend becomes less consistent as reasoning complexity increases to Orders 2-4, where certain non-reasoning models begin to demonstrate comparable or superior performance. A stark example can be seen in the DeepSeek family on 4th-order tasks: the non-reasoning DeepSeek-V3 achieves a robust score of 0.608, while the performance of the reasoning-focused DeepSeek-R1 collapses to 0.196. This susceptibility to failure under high complexity is even more pronounced in models like Grok-3-mini, whose accuracy plummets to 0.197 as early as Order 2, suggesting that some reasoning models are susceptible to reasoning collapse when confronted with complex scenarios. A key exception to this pattern is

Claude-Sonnet-4, which sustains an impressively high accuracy of 0.721 even at the 4th-order. A more in-depth analysis of these behaviors is in Section 4.2.1.

### 4.1.2 REASONING'S BENEFITS VARY ACROSS TAXONOMY

We also study the influence of taxonomy on the performance of LLMs, as shown in Table 2. A consistent advantage for reasoning models emerges in categories requiring the inference of structured, propositional attitudes. For example, in Belief, Intention, and the more challenging Knowledge categories, advanced reasoning models like Claude-Sonnet-4 and GPT-o3 consistently outperform the non-reasoning models, suggesting explicit thinking path is particularly effective for tracking cognitive states. However, this performance gap diminishes when assessing some mental states. Most notably, in the Desire category, the non-reasoning Qwen3-32B is tied for the top score at 0.689. These results reveal that the improvements from current reasoning capabilities are selective, rather than universal. While they may enhance some aspects of ToM, they fail to provide a discernible advantage in challenging inferences, such as desire.

## 4.2 WHY REASONING FAILS TO OUTPERFORM?

### 4.2.1 RESPONSE LENGTH AS A SIGNATURE OF FAILURE

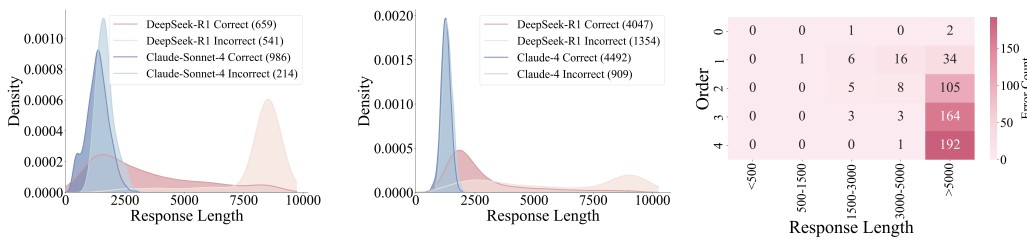

(a) Response analysis on HiToM   (b) Response analysis on ToMATO   (c) Orders and lengths in HiToM

Figure 1: The distribution of the length and correctness of reasoning model responses.

To understand how reasoning strategy changes between successful and failed attempts, we analyze the response length distribution for DeepSeek-R1. The results reveal a striking pattern, particularly on the HiToM benchmark. As shown in Figure 1a, the errors made by DeepSeek-R1 on this benchmark predominantly appear in a high-response-length region, forming a massive peak around 8,000 to 10,000 characters. This contrasts sharply with the higher-performing Claude-Sonnet-4, whose responses are consistently concise. However, this extreme pattern is mitigated on the ToMATO and ToMBench (Appendix B.4 ) benchmarks. While a distinction between the length of correct and incorrect responses for DeepSeek-R1 still exists on ToMATO, the separation is less pronounced, the error count is lower, and the distributions are more dispersed. The reason for this difference becomes clear when we analyze the source of these failures. Figure 1c, a heatmap of DeepSeek-R1's errors on HiToM, shows that the higher the task complexity, the more likely the model is to produce long and erroneous responses. While HiToM scales to the highly challenging Order 4, ToMATO peaks at the less demanding Order 2. This suggests reasoning failure correlates with response length, especially on complex tasks.

### 4.2.2 HIGHER REASONING EFFORT DOES NOT LEAD TO BETTER PERFORMANCE

Our experiments on the influence of reasoning effort also provide strong evidence for the negative impact of slow thinking. On the complex, higher-order HiToM benchmark, we observe a clear inverse relationship between effort and accuracy. As illustrated in Figure 2a, the performance of the GPT-o3 model drops substantially from a high of 0.838 at the lowest effort level to 0.693 at the highest. This negative correlation is

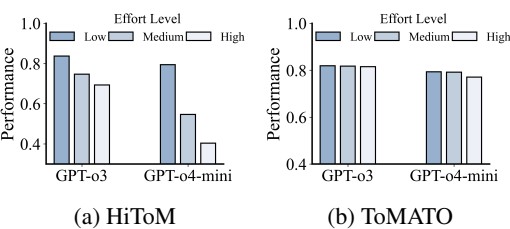

(a) HiToM   (b) ToMATO

Figure 2: Model performance with various reasoning efforts on benchmarks.

weakened on the less complex ToMATO benchmark (Figure 2b) and ToMBench (Appendix B.6). We found that varying the reasoning effort had a negligible impact on final accuracy. This divergence between two benchmarks demonstrates that the slow thinking failure is triggered by the high cognitive load of complex tasks.

> 💡 **Takeaway 2:** On complex ToM tasks, heavy reasoning expenditure, suggested by more thinking tokens or controlled by reasoning effort parameter, is associated with performance degradation rather than improvement.

### 4.2.3 COMPLEMENTARY STRENGTHS OF REASONING AND NON-REASONING

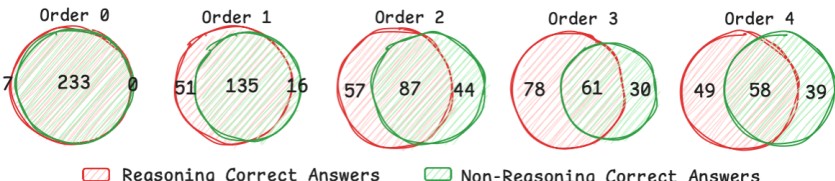

Figure 3: Overlap between reasoning and non-reasoning models' correct answers.

To investigate how task complexity impacts reasoning versus non-reasoning models, we use Qwen3-32B as a case study. We perform a granular comparison of decision outcomes between the standard Qwen3-32B and the Qwen3-32B-Reasoning variant on the HiToM benchmark to illuminate their evolving roles. The results in Figure 3 reveal a clear pattern: the models' capabilities are redundant in simple scenarios but become highly complementary as task complexity increases. In low-complexity scenarios like Order 0 and Order 1, the models are largely in agreement, correctly answering a shared set of 233 and 135 samples, respectively. However, a significant divergence emerges at Order 2, where their complementary strengths become apparent with 87 overlapping correct answers but a combined 101 distinct correct answers where only one model succeeded. This trend culminates at the highest complexity, Order 4, where the Reasoning model uniquely solves 49 samples and the Non-Reasoning model uniquely solves 39, while they only agree on 58. This substantial non-overlapping success clearly demonstrates that each model possesses unique problem-solving abilities that are crucial for tackling complex ToM challenges. These complementary strengths imply that neither pure reasoning nor non-reasoning models alone can achieve optimal results. Instead, an adaptive strategy that dynamically selects the appropriate reasoning budget based on task complexity offers a more promising path to superior performance. More experimental results are shown in Appendix B.7.

### 4.2.4 MODERATE THINKING IS HELPFUL

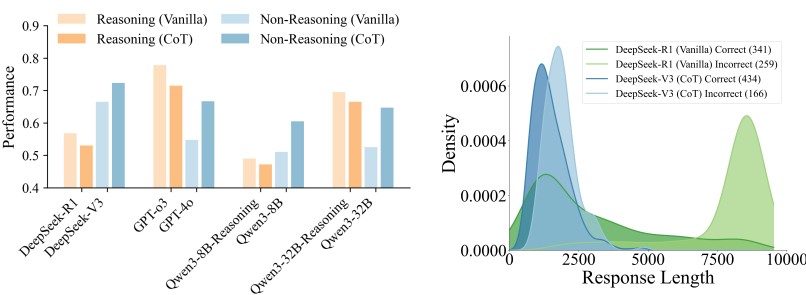

(a) Model performance with CoT prompting   (b) Response lengths and correctness

Figure 4: Model performance with CoT prompting.

**Study of CoT Prompt.** To determine whether a generic reasoning instruction like rule-based Chain-of-Thought (CoT) would help non-reasoning models and avoid triggering reasoning models' failure modes, we conduct experiments to study the usefulness of CoT prompt on the complex HiToM benchmark. Figure 4a shows that CoT prompting provides a significant boost to non-reasoning models. For example, the performance of GPT-4o increases substantially from 0.547 to 0.667. Furthermore,

our visualization of the response length distribution in Figure 4b also shows an interesting pattern. DeepSeek-V3 with CoT engages in moderate thinking, a process that improves its performance while successfully avoiding the pitfalls of prolonged deliberation and reasoning collapse. This balanced approach elevates the model's performance to nearly the level of the advanced GPT-o3. This suggests that while reasoning is inherently beneficial, the specific cognitive processes of current reasoning models may be flawed or misaligned with the unique demands of ToM tasks, which leads to the observed failure of reasoning. Finally, a minor finding shows that CoT is counterproductive for reasoning models, leading to a degradation in their performance.

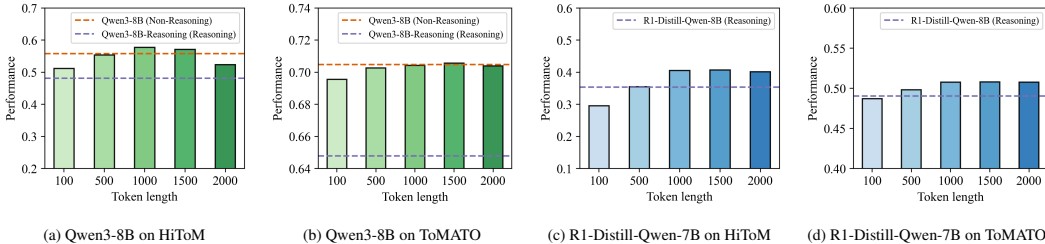

| (a) Qwen3-8B on HiToM | (b) Qwen3-8B on ToMATO | (c) R1-Distill-Qwen-7B on HiToM | (d) R1-Distill-Qwen-7B on ToMATO |

Figure 5: Performance comparison under different token length limitations. The dash lines show the original model performance without token limitation.

**Study of Token Control.** To further investigate the correlation between token length and performance, we conduct an experiment by strictly limiting the thinking tokens. Specifically, we set various thresholds for the maximum thinking length. When the length surpasses the threshold, we terminate the thinking process and append $</think>$ to the internal thought to force the LLM to derive an answer. We report the performance in Figure 5a. We observe that on both HiToM and ToMATO, the optimal token length is approximately 1500. With this constraint, the performance surpasses both unconstrained reasoning and non-reasoning baselines. For example, Qwen3-8B with a 1500-token limit achieves the performance of 0.7056, whereas the standard Qwen3-8B and Qwen3-8B-Reasoning achieve 0.7047 and 0.6478, respectively. This suggests that while reasoning is beneficial, it requires control to remain both effective and efficient.

> 💡 **Takeaway 3:** Moderate and adaptive reasoning outperforms unconstrained slow thinking or no thinking, implying the needs for thinking strategy enhancement in ToM tasks.

### 4.2.5 OPTION HINDERS SUCCESSFUL REASONING

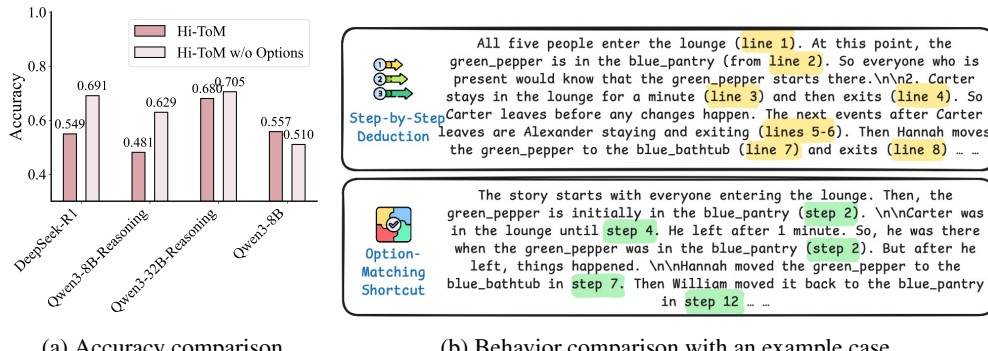

| (a) Accuracy comparison | (b) Behavior comparison with an example case |

Figure 6: The comparison of model performance when options are provided or not.

To investigate the influence of options on reasoning, we modified the HiToM benchmark by removing multiple-choice candidates, requiring models to extract answers directly. HiToM was selected specifically because its answers are explicitly extractive, ensuring unambiguous open-ended evaluation. As shown in Figure 6a, removing options yields substantial improvements for reasoning models. For example, DeepSeek-R1 surges from 0.549 to 0.691, and Qwen3-8B-Reasoning improves from 0.481 to 0.629. In contrast, the non-reasoning Qwen3-8B drops from 0.557 to 0.510. Qualitative

analysis (Figure 6b) explains this divergence: explicit options trigger an "option matching heuristic." When presented with options, reasoning models often abandon linear deduction, instead engaging in a chaotic search to find superficial justifications for potential choices. Conversely, the option-free format compels them to perform structured, step-by-step deduction. These results suggest that explicit candidates short-circuit deduction, causing reasoning models to anchor on shallow overlaps. Meanwhile, non-reasoning baselines depend heavily on choice-level signals and lose support when those signals are removed.

> 💡 **Takeaway 4:** Explicit options short-circuit reasoning, causing models to prioritize superficial matching over deduction.

### 4.3 PROBING REASONING FAILURE AND OPTIMIZATION STRATEGIES IN ToM

As we have identified a series of problems in ToM reasoning, we develop two optimization strategies to further verify and mitigate the problems in this section. First, we leverage Slow-to-Fast reasoning to investigate the potential of an adaptive cognitive strategy. Building on it, we design Think-to-Match to deeply study the influence of the options on reasoning.

#### 4.3.1 SLOW-TO-FAST REASONING

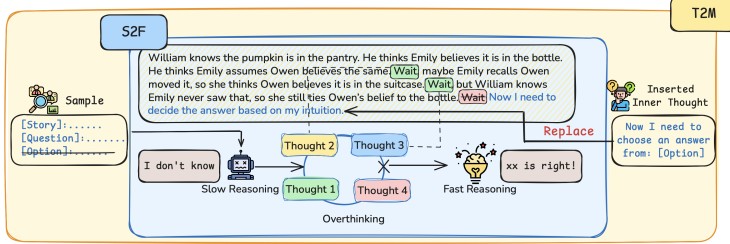

Figure 7: Overview of the Slow-to-Fast (S2F) and Think-to-Match (T2M) techniques.

The existing works mark *"wait"* as a proxy for extended deliberation. In reasoning models, such markers typically trigger additional verification or strategy switches, thereby lengthening the reasoning trajectory (Guo et al., 2025). To avoid prolonged slow thinking, inspired by previous research (Zhang et al., 2025), S2F uses the frequency of *"wait"* as a direct trigger for our dynamic intervention When its count exceeds a threshold, the overlong reasoning path tends to be unproductive in ToM tasks. Thus, we terminate the slow thinking process and compel the model to switch to fast thinking for an intuitive answer.

The results in Table 3 reveal that the effectiveness of S2F intervention is directly correlated with task complexity. The benefits are most pronounced on the complex, higher-order benchmark HiToM. For instance, the R1-Distill-Qwen-32B model's performance surges from 0.571 to 0.701. However, on the 1st-order ToMBench, the intervention has a negligible impact across all models. These results offers a two-fold conclusion. First, the performance gains on HiToM demonstrate that the S2F strategy can successfully mitigate the problem of redundant reasoning. It implies that most of the reasoning models cannot successfully determine the reasoning efforts for ToM questions and the problem is more severe in complex questions. They do not lack ToM capabilities, but their wrong reasoning strategy damages performance. Furthermore, though our results verifies that moderate and adaptive thinking efforts can improve ToM performance, how to achieve the dynamic adjustment for thinking efforts requires more future research. We notice that the model performance improvement led by S2F on ToMATO and ToMBench is less salient than the improvement on HiToM. As discussed in Section 4.2.1, the detrimental effects of slow thinking are less pronounced on simpler benchmarks, such as ToMATO. We suspect that the benefits gained from inhibiting slow thinking are counterbalanced by the disruption to the natural reasoning process, resulting in negligible overall improvement or even a slight decline in performance. It suggests that the adaptive reasoning should be achieved by considering both deliberation efforts and question complexity, implying more future research on genuine ToM reasoning capability.

Table 3: S2F performance across various reasoning models.

| Benchmark | Qwen3-8B | | Qwen3-32B | | R1-Distill-Qwen-7B | | R1-Distill-Qwen-32B | | R1-Distill-Llama-8B | |
|---|---|---|---|---|---|---|---|---|---|---|
| | Vanilla | S2F | Vanilla | S2F | Vanilla | S2F | Vanilla | S2F | Vanilla | S2F |
| HiToM | 0.481 | 0.557 (+15.8%) | 0.680 | 0.682 (+0.3%) | 0.353 | 0.397 (+12.5%) | 0.571 | 0.701 (+22.8%) | 0.396 | 0.451 (+13.9%) |
| ToMATO | 0.648 | 0.700 (+8.0%) | 0.714 | 0.724 (+1.4%) | 0.490 | 0.505 (+3.1%) | 0.706 | 0.708 (+0.3%) | 0.586 | 0.578 (-1.4%) |
| ToMBench | 0.729 | 0.731 (+0.3%) | 0.775 | 0.777 (+0.3%) | 0.559 | 0.560 (+0.2%) | 0.773 | 0.769 (-0.5%) | 0.655 | 0.626 (-4.4%) |

### 4.3.2 THINK-TO-MATCH

Building on S2F, we introduce T2M to prevent option matching shortcuts. In the thinking phase, T2M removes answer options, compelling the model to perform first-principles reasoning constrained only by the S2F monitor. In the subsequent matching phase, options are reintroduced, prompting the model to align its generated deduction with the candidates. This approach ensures decisions are grounded in independent reasoning rather than superficial heuristics. As shown in Figure 8, T2M significantly improves performance on HiToM, verifying that eliminating shortcuts unleashes the ToM capabilities of LRMs.

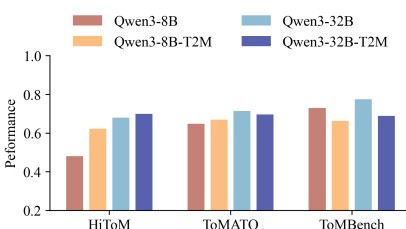

Figure 8: T2M performance.

Furthermore, we learn an important lesson on ToM benchmark design from T2M's ineffectiveness on ToMBench. Cases in Figure 9 reveal that the wrong results by T2M often follow the pattern: when the options do not match the initial reasoning direction, the answer can be wrong. This issue is related to the benchmark designs. In ToMBench, questions are often more open-ended, such as inferring potential activities that a person might invite another to do. These questions can have a large answer space, meaning that models can take various thinking directions to generate many more potentially correct answers than those provided as options. As a result, without showing potential options in advance, T2M can lead to mismatch between the thinking directions and the provided options, further resulting in a wrong answer. With the lens of T2M, a considerable challenge in ToM evaluation is revealed. In a benchmark, questions with options can greatly restrict the search space for answers and thus lead to undesired option matching shortcuts. The performance growth led by these option matching behaviors is risky, as the ToM capability might be over-estimated. On the other hand, if we only present the open-ended questions to models, the evaluation can be tricky, as the reasoning process may converge in many directions, leading to challenges of assessing them consistently.

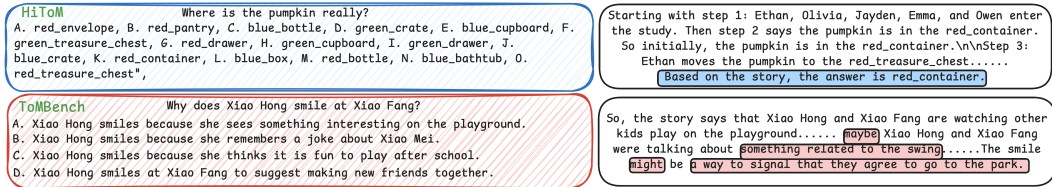

Figure 9: Cases to illustrate the different performance of T2M on HiToM and ToMBench.

### 4.4 DISCUSSION

Moving beyond the above experimental results, we derive key insights from the reasoning failure and provide inspiration for future directions in ToM reasoning method design.

### 4.4.1 EMPIRICAL INSIGHTS FROM REASONING FAILURE.

**Slow Thinking Correlates with Reasoning Failures.** We find that slow thinking in ToM tasks is strongly correlated with failure from the analysis in Section 4.2.1 and 4.2.2. This occurs because the inherent ambiguity of ToM tasks makes extended deliberation a liability. A model can become trapped in a divergent search for plausible-but-incorrect interpretations, leading to counterproductive loops of self-correction that derail the entire reasoning process. Our analysis in Section 4.2.4 and 4.3.1 also supports this insight by examining the impact of constraining thinking tokens.

**Reasoning Takes Option Matching Shortcut.** The analysis in Section 4.2.5 demonstrates when the multiple-choice options are presented, reasoning models often abandon genuine deduction for a brittle, reverse-lookup process to justify each choice, making them susceptible to plausible distractors. Additionally, our T2M results in Section 4.3.2 indicate that enforcing a 'think-then-choose' strategy effectively improves performance, thereby validating our diagnosis of this issue.

**Moderate and Adaptive Reasoning Benefits Performance.** From Section 4.2.4, 4.2.3, and 4.3.1, we demonstrate that constraining reasoning length prevents cognitive collapse, while the distinct success patterns on complex tasks highlight the necessity of dynamically adapting between reasoning and non-reasoning paradigms to maximize performance.

### 4.4.2 FROM FORMAL REASONING TO ToM REASONING.

**Reasoning Effort Aids Formal Reasoning but Impairs ToM Reasoning.** ToM contrasts with formal reasoning by exhibiting a bi-phasic profile (Figure 1a): correct answers are concise, while errors cluster in high-length regions. This indicates a "reasoning collapse," where prolonged deliberation proves counterproductive. Traditionally, reasoning models show a distinct performance gap with non-reasoning models in formal reasoning. For example, DeepSeek-R1 performs clearly better than DeepSeek-V3 (e.g., math, code) (Guo et al., 2025). However, reasoning models fail to demonstrate dominated advantages on ToM benchmarks. We attribute this to ToM's weakly verifiable nature. Unlike formal reasoning, where checkable intermediate states allow evidence to accumulate, extended ToM reasoning tends to amplify noise and induce perspective drift, often overwriting correct initial intuitions.

**Multiple Choice Helps Formal Reasoning, Hurts ToM Reasoning.** A second critical distinction lies in the impact of multiple-choice options. As we demonstrated in Figure 6a, for ToM reasoning tasks, the presence of an option set often degrades reasoning-model performance: models may adopt an option matching shortcut, becoming vulnerable to distractors. By contrast, in formal reasoning domains, a multiple-choice format can help by constraining the hypothesis space and providing clear targets for verification, often improving performance (Raman et al., 2025). These divergent outcomes show that ToM and formal reasoning are qualitatively distinct problem classes requiring different capabilities and strategies.

### 4.4.3 TOWARDS ToM REASONING IMPROVEMENT

**A System 1 & System 2 Perspective.** Our findings suggest that progress on ToM reasoning can be fruitfully framed by dual-process theory, where non-reasoning models approximate an intuitive System 1 and reasoning models approximate a deliberative System 2 (Ziabari et al., 2025; Sui et al., 2025). This view reconciles our results: Section 4.2.3 shows complementary strengths that neither system suffices universally. Section 4.2.4 shows that a System 1-like model can be guided with simple CoT prompt to outperform System 2-like deliberate thinking. A key limitation is that current systems are typically run in a fixed mode per instance, rather than adapting effort to instance difficulty. Thus the goal is not merely building a stronger System 2, but integrating adaptive strategy selection. Future work can explore hybrids (e.g., a System 1 proposer with a System 2 verifier) and training that rewards choosing the minimal effective reasoning path. By shifting the target from thinking more to knowing how and when to think, we move toward more robust, human-like ToM reasoning.

## 5 CONCLUSION

In this paper, we investigate whether LRMs can enhance ToM, and find that their direct application is often ineffective and sometimes detrimental. Our systematic comparison reveals that reasoning models consistently fail to outperform their non-reasoning counterparts due to two primary failure reasons: reasoning collapse, where prolonged deliberation backfires, and a reliance on brittle option-matching shortcuts. These findings highlight a fundamental divergence between the requirements of formal and social reasoning, showing that strategies successful in logic-based tasks become liabilities in ambiguous social contexts. This implies that improving ToM is not about simply scaling existing analytical methods but requires developing unique capabilities. Our preliminary explorations with Slow-to-Fast (S2F) reasoning and the Think-to-Match (T2M) analysis provide initial steps in this new direction toward more adaptive and efficient social intelligence.

ETHICS STATEMENT

We acknowledge that ToM capabilities in LLMs carry potential risks, including manipulative use. However, we argue that thoroughly understanding these mechanisms is essential for building effective defenses. Our work serves as a critical diagnosis rather than a capability advancement. By revealing that some reasoning models often fail due to "slow thinking collapse" and "option matching shortcuts", we provide a necessary reality check to prevent over-trust in current systems. This empirical understanding of model limitations is a prerequisite for developing transparent and controllable AI systems.

REPRODUCIBILITY STATEMENT

To facilitate reproducibility, we provide our source code at `https://anonymous.4open.science/r/ToM-Reasoning-312B/`. We also provide our hyperparameters for LLMs in Section 3.1 and the pseudo-code of methods in Section C.1.

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

## A  LLM USAGE STATEMENT

We used LLMs (e.g., ChatGPT) only for grammar and wording edits.

## B  EXPERIMENTAL RESULTS

### B.1  OVERALL COMPARISON

A comprehensive evaluation across the three ToM benchmarks in Figure 10 shows a heterogeneous performance landscape. Claude-Sonnet-4 sets the state of the art, leading all three benchmarks with HiToM 0.8217, ToMATO 0.8317, and ToMBench 0.8315. GPT-o3 performs near this level with consistently strong results. Most other models vary widely by benchmark, indicating strengths that are dependent on the benchmark rather than uniform capability.

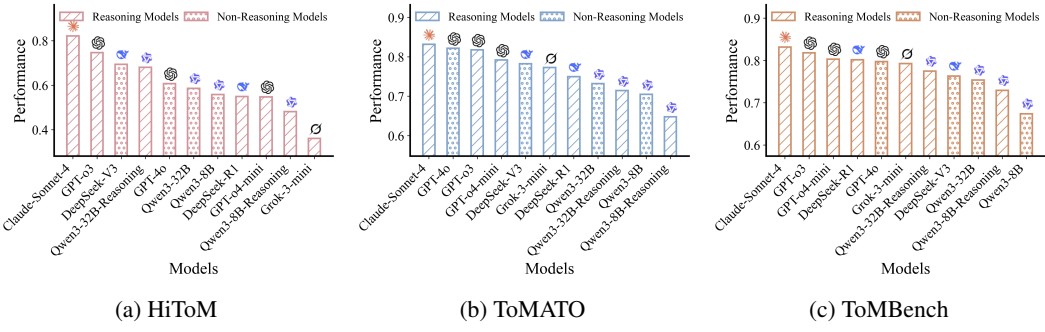

(a) HiToM          (b) ToMATO          (c) ToMBench

Figure 10: Overall comparison on three ToM benchmarks.

Table 4: General, cross-benchmark error taxonomy with simplified definitions and examples that highlight faulty model thinking. Each row uses a distinct highlight color.

| Error Type | Description | Example |
|---|---|---|
| **Evidence Grounding Error** | Prediction is not grounded on the evidence or cites an entity that is not supported by the given context. | *Thinking:* "Choose `red_basket`; it sounds plausible." 
 *Fault:* `red_basket` is not in the evidence set; no justification links it to the task options. |
| **State Tracking Error** | Timeline or observability is tracked incorrectly. | *Thinking:* "The last move puts it in X, so Owen believes X." 
 *Fault:* Owen had already left before the last move; belief should freeze at his last observation. |
| **Perspective Attribution Error** | Answers the wrong subject or wrong mental construct. | *Thinking:* "I (A) saw it in Y; therefore B thinks Y." 
 *Fault:* Egocentric projection; the question asks for A's belief about B's belief, not A's own knowledge. |
| **Discourse Misinterpretation** | Speech acts or rhetorical cues are treated as factual updates. | *Thinking:* "He *claimed* it is in X; so it is in X and others now know X." 
 *Fault:* Claims do not change world state or shared knowledge without corroboration. |
| **Commonsense & Causal Error** | Over-applies a generic script or inserts unsupported causality not warranted by context. | *Thinking:* "This looks like a negotiation; the goal is to persuade, so the counterpart is cautious." 
 *Fault:* Injects a persuasion script and causal story absent from the evidence; mislabels the speaker's stance. |

## B.2 IDENTIFYING REASONING ERROR TYPES

To understand why models fail, we move from quantitative scoring to a qualitative analysis of their reasoning processes. By manually inspecting the chain-of-thought outputs for incorrect predictions, we identified five recurring and distinct categories of reasoning failures. These categories, which form a general, cross-benchmark error taxonomy, are defined and exemplified in Table 4.

**Evidence Grounding Errors.** This is one of the most fundamental types of failure, where the model's reasoning path deviates from the provided context. An Evidence Grounding Error occurs when a model bases its conclusion on information that is not present in the evidence or makes a decision by citing an entity that cannot be factually supported by the scenario. As the example in Table 4 shows, this can manifest as the model choosing an option because it "sounds plausible" rather than because it is logically derived from the given facts. This error type highlights a critical weakness in a model's ability to strictly adhere to its input context, often resorting to unverified assumptions or hallucinated details.

**State Tracking Error.** This error reveals a model's difficulty with temporal and observational logic. A State Tracking Error happens when the model incorrectly processes the timeline of events or fails to account for an agent's limited perspective. A common failure mode is updating an agent's belief based on an event that the agent did not witness. For instance, a model might incorrectly conclude that "Owen believes X" because the object was moved to location X, while failing to register that Owen had already left the room and could not have observed the move. This points to a deficient mechanism for maintaining and freezing the mental state of different agents at specific points in time.

**Perspective Attribution Error.** This error is a classic failure of Theory of Mind, where the model fails to correctly simulate another agent's perspective and instead defaults to its own. A Perspective Attribution Error occurs when the model answers from the wrong point of view—often its own "all-seeing" one—or confuses whose mental state is being queried. This frequently manifests as egocentric projection, where the model imputes its own knowledge onto an agent (e.g., "I, the model, know it is in Y, therefore Agent B must think it is in Y"). This shows a breakdown in handling nested beliefs and maintaining the crucial distinction between objective reality and an agent's subjective perception.

**Discourse Misinterpretation.** This category of error highlights a model's lack of pragmatic understanding in social communication. A Discourse Misinterpretation occurs when the model treats non-factual speech acts—such as claims, questions, jokes, or rhetorical statements—as literal updates to the world state or shared knowledge. For example, a model might incorrectly assume that because a character claimed an object was in a certain location, the object is now factually in that location and all other characters are aware of this. This reveals a naive, literal interpretation of language that misses the social nuances and reliability judgments inherent in human discourse.

**Commonsense & Causal Error.** This error type involves the misapplication of world knowledge or the fabrication of unsupported causal links. A Commonsense & Causal Error happens when a model imposes a generic script or schema onto a situation where it does not fit, or when it invents a cause-and-effect relationship that is not warranted by the evidence. For example, a model might incorrectly classify a simple information exchange as a "negotiation," thereby misinterpreting the agents' stances and intentions. This shows that while models possess vast commonsense knowledge, they struggle to apply it appropriately and can over-generalize, leading to a distorted understanding of the specific context.

## B.3 DETAILED RESULTS ON TOMATO

Detailed results on ToMATO across different reasoning order and taxonomy is provided in Table 5

Table 5: ToMATO fine-grained accuracy by order and taxonomy

| Model | 1st-Order | | | | | 2nd-Order | | | | | All | | | | |
|---|---|---|---|---|---|---|---|---|---|---|---|---|---|---|---|
| | belief | desire | emotion | intention | knowledge | belief | desire | emotion | intention | knowledge | belief | desire | emotion | intention | knowledge |
| Qwen3-32B | 0.773 | 0.848 | 0.772 | 0.802 | 0.747 | 0.599 | 0.731 | 0.727 | 0.658 | 0.680 | 0.685 | 0.782 | 0.749 | 0.735 | 0.714 |
| Qwen3-32B-Reasoning | 0.773 | 0.859 | 0.788 | 0.751 | 0.696 | 0.586 | 0.724 | 0.713 | 0.617 | 0.663 | 0.679 | 0.783 | 0.750 | 0.689 | 0.680 |
| Qwen3-8B | 0.717 | 0.830 | 0.760 | 0.810 | 0.686 | 0.595 | 0.703 | 0.705 | 0.660 | 0.606 | 0.656 | 0.759 | 0.732 | 0.741 | 0.646 |
| Qwen3-8B-Reasoning | 0.719 | 0.839 | 0.735 | 0.750 | 0.644 | 0.493 | 0.570 | 0.677 | 0.518 | 0.572 | 0.605 | 0.689 | 0.706 | 0.642 | 0.608 |
| Claude-Sonnet-4 | 0.891 | 0.894 | 0.840 | 0.844 | 0.837 | 0.785 | 0.857 | 0.846 | 0.726 | 0.803 | 0.838 | 0.873 | 0.843 | 0.789 | 0.820 |
| Grok-3-mini | 0.840 | 0.859 | 0.820 | 0.827 | 0.747 | 0.699 | 0.788 | 0.806 | 0.668 | 0.691 | 0.769 | 0.820 | 0.813 | 0.753 | 0.719 |
| GPT-4o | 0.844 | 0.901 | 0.848 | 0.859 | 0.808 | 0.767 | 0.836 | 0.812 | 0.744 | 0.807 | 0.805 | 0.864 | 0.830 | 0.805 | 0.807 |
| GPT-o3 | 0.866 | 0.874 | 0.867 | 0.859 | 0.814 | 0.737 | 0.827 | 0.864 | 0.720 | 0.758 | 0.801 | 0.848 | 0.865 | 0.795 | 0.786 |
| GPT-o4-mini | 0.851 | 0.879 | 0.818 | 0.849 | 0.808 | 0.695 | 0.806 | 0.774 | 0.672 | 0.772 | 0.773 | 0.838 | 0.796 | 0.767 | 0.790 |
| DeepSeek-R1 | 0.789 | 0.874 | 0.786 | 0.827 | 0.720 | 0.670 | 0.726 | 0.758 | 0.672 | 0.693 | 0.729 | 0.791 | 0.772 | 0.755 | 0.706 |
| DeepSeek-V3 | 0.826 | 0.863 | 0.804 | 0.832 | 0.751 | 0.706 | 0.794 | 0.796 | 0.713 | 0.750 | 0.766 | 0.824 | 0.800 | 0.777 | 0.750 |

## B.4 RESPONSE LENGTH DISTRIBUTION

The response length distributions for the models are presented for each benchmark: HiToM in Figure 11, ToMATO in Figure 12, and ToMBench in Figure 13. A clear trend emerges when comparing these distributions in order of task complexity. The distinct pattern of failure we have identified—where errors cluster in an extremely high response-length region—is most pronounced on the most complex benchmark, HiToM. This effect is noticeably mitigated on ToMATO and is least apparent on ToMBench. This progression provides strong corroborating evidence for our hypothesis: the counterproductive slow thinking that leads to reasoning collapse is a failure mode that is specifically triggered and amplified by high task complexity.

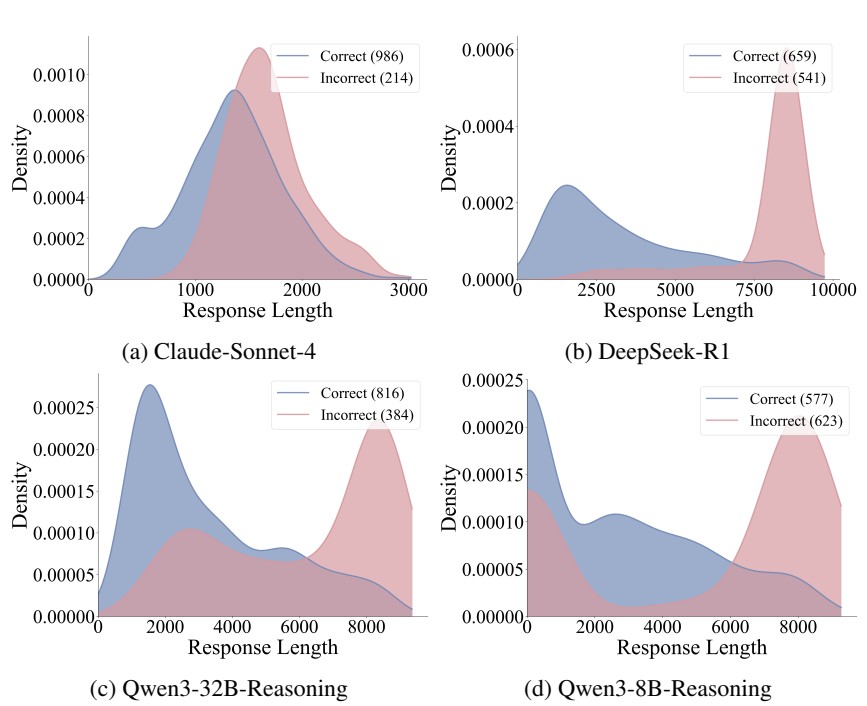

Figure 11: Response Length Distribution on HiToM.

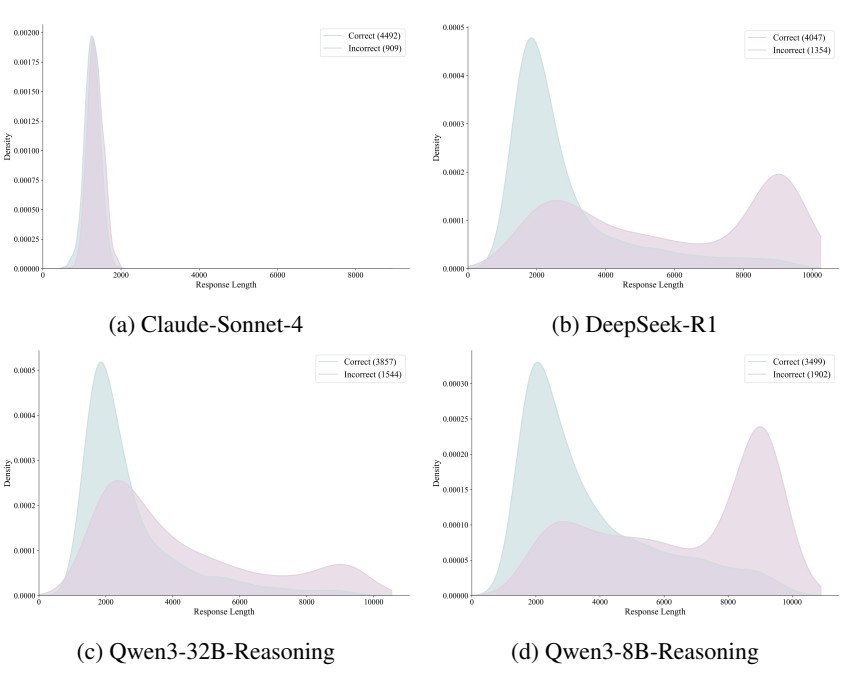

Figure 12: Response Length Distribution on ToMATO.

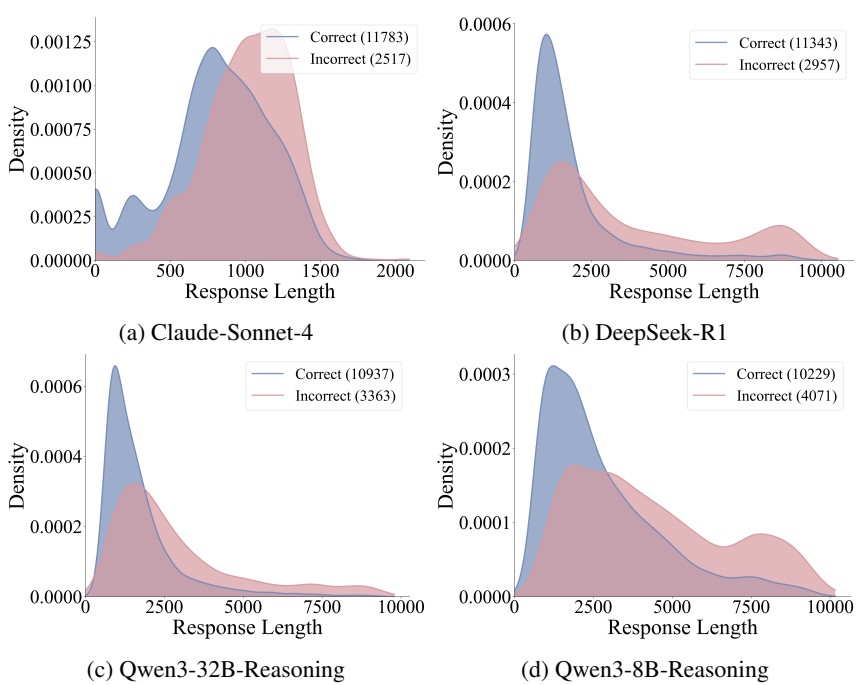

Figure 13: Response Length Distribution on ToMBench.

## B.5 ORDER AND LENGTH

We provide the heatmaps of incorrect answers on different orders and response lengths on HiToM in Figure 14.

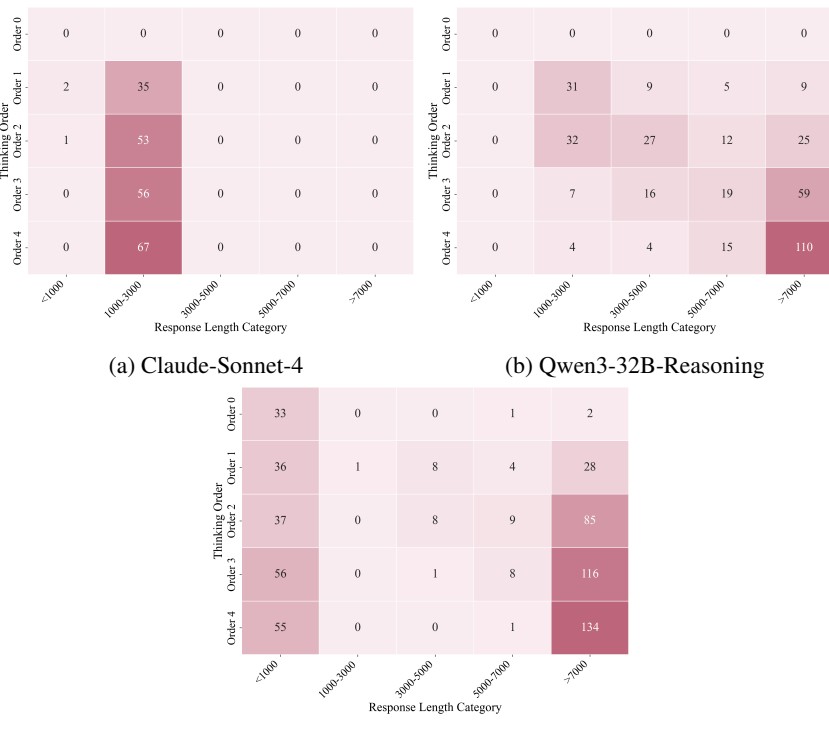

(a) Claude-Sonnet-4    (b) Qwen3-32B-Reasoning

(c) Qwen3-8B-Reasoning

Figure 14: Order and Length on HiToM

## B.6 REASONING EFFORT

We provide the results of different reasoning efforts on ToMBench in Figure 15. The findings are similar to those on ToMATO: increasing the reasoning effort does not lead to a significant change in performance. This reinforces our conclusion that the detrimental effects of slow thinking are specifically triggered by high task complexity. On less complex benchmarks like ToMBench and ToMATO, the reasoning collapse failure mode is not induced, and therefore, additional computational effort is neither beneficial nor harmful.

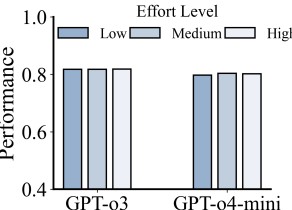

Figure 15: Resoning effort on ToMBench.

## B.7 CORRECT ANSWER OVERLAP

We provide the results of correct answer overlap on HiToM in Figure 16 and on ToMATO in Figure 17. They aligns with the observation in Figure 3, where the overlap between correct answers of

reasoning and non-reasoning models in the same family grows when the order is higher. The results imply the complementary advantages of two types of models.

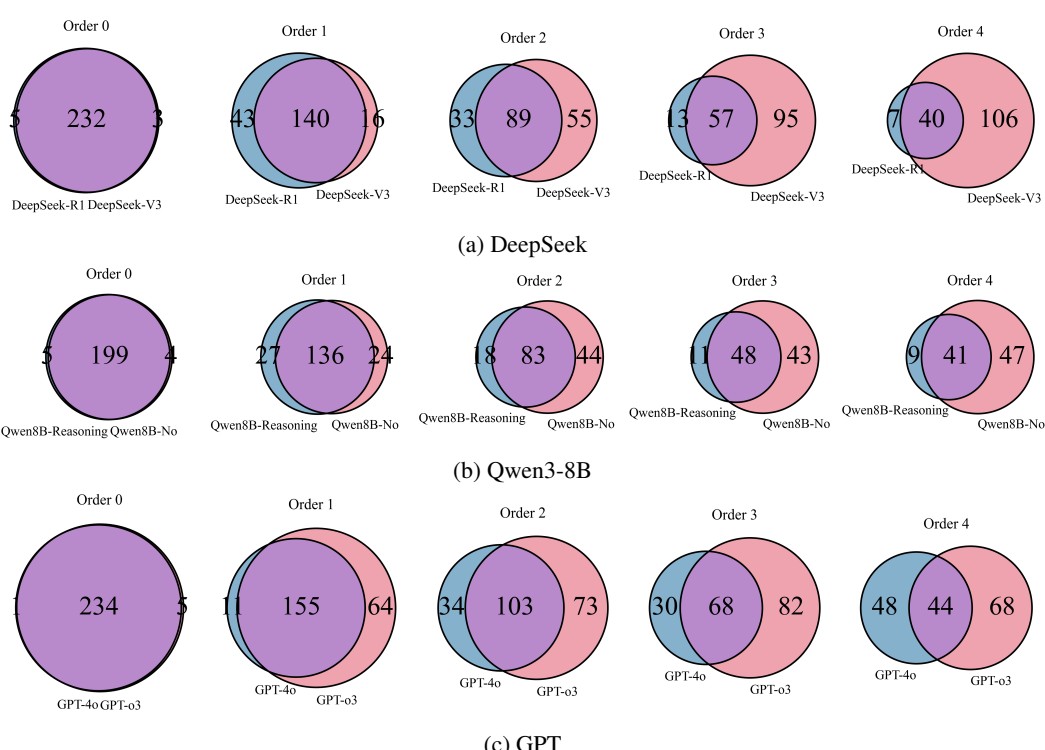

Figure 16: Correct Answer Overlap on HiToM

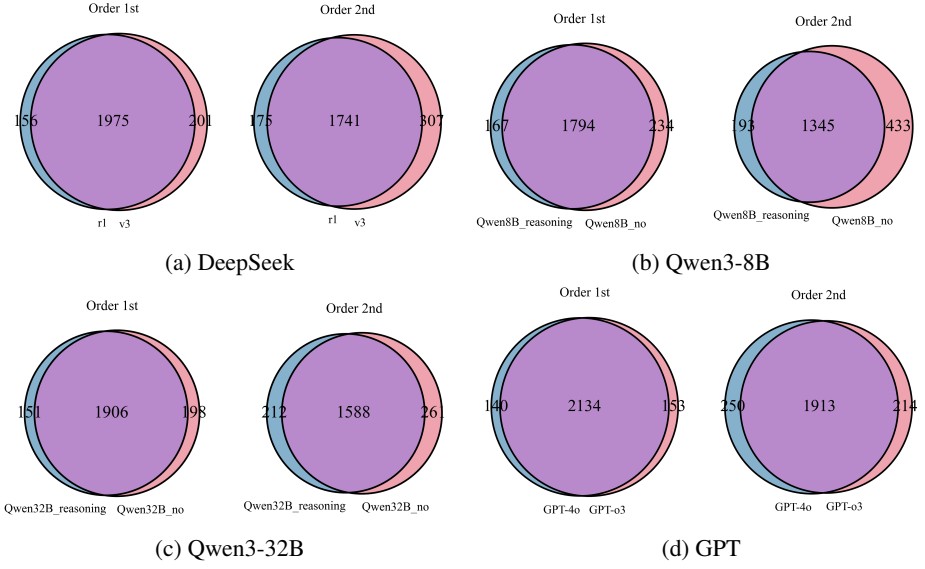

Figure 17: Correct Answer Overlap on ToMATO

# C METHOD

## C.1 PSEDUO-CODE

We provide pseudo-code of S2F and T2M in Algorithm 1 and 2 respectively for reproducibility. Specifically, we set the threshold of "wait" count to 3 in our experiments. As S2F intervention requires token-level generation control, we conduct these experiments on open-source models. To analyse comprehensively, we use Qwen3 models and introduce two R1-Distill-Qwen variants.

---

**Algorithm 1** Slow-to-Fast Reasoning

---

1: **Input:** Prompt $x$, LLM $f_\theta$, threshold $\tau$, target token $w =$ "wait", preset sentence $S_{\text{ins}}$, max length $L_{\max}$
2: **Output:** Generated text $Y$
3: $Y \leftarrow$ ""; $c \leftarrow 0$; finished $\leftarrow$ **false**
4: **while** not finished **do**
5:     $p(\cdot) \leftarrow f_\theta(x \oplus Y)$
6:     $t^\star \leftarrow \arg\max_t p(t)$
7:     **if** $c \geq \tau - 1$ and $t^\star = w$ **then**
8:         $Y \leftarrow Y \oplus S_{\text{ins}}$
9:         $c \leftarrow 0$
10:    **else**
11:       sample $t \sim \text{Decode}(p(\cdot))$
12:       $Y \leftarrow Y \oplus t$
13:       **if** $t = w$ **then**
14:         $c \leftarrow c + 1$
15:       **end if**
16:       **if** $t$ is EOS or length$(Y) \geq L_{\max}$ **then**
17:         finished $\leftarrow$ **true**
18:       **end if**
19:    **end if**
20: **end while**
21: **return** $Y$

---

---

**Algorithm 2** Think-to-Match

---

1: **Inputs:** base prompt $x_{\text{base}}$ (prompt without options), options string $O$, LLM $f_\theta$, threshold $\tau$, target token $w =$ "wait", preset sentence $S_{\text{ins}}$, max length $L_{\max}$
2: **Output:** Generated text $Y$
3: $S_{\text{full}} \leftarrow S_{\text{ins}} \oplus O$
4: $Y \leftarrow$ ""; $c \leftarrow 0$; finished $\leftarrow$ **false**; inserted $\leftarrow$ **false**
5: **while not** finished **do**
6:     $p(\cdot) \leftarrow f_\theta(x_{\text{base}} \oplus Y)$
7:     $t^\star \leftarrow \arg\max_t p(t)$
8:     **if** $(c \geq \tau - 1)$ **and** $(t^\star = w)$ **and not** inserted **then**
9:         $Y \leftarrow Y \oplus S_{\text{full}}$
10:         $c \leftarrow 0$; inserted $\leftarrow$ **true**
11:         **continue**
12:     **end if**
13:     **if** $(t^\star =</think>)$ **and not** inserted **then**
14:         $Y \leftarrow Y \oplus S_{\text{full}}$
15:         $c \leftarrow 0$; inserted $\leftarrow$ **true**
16:         **continue**
17:     **end if**
18:     sample $t \sim \text{Decode}(p(\cdot))$
19:     $Y \leftarrow Y \oplus t$
20:     **if** $t = w$ **then**
21:         $c \leftarrow c + 1$
22:     **end if**
23:     **if** $t = \langle \text{END} \rangle$ **or** $\text{length}(Y) \geq L_{\max}$ **then**
24:         finished $\leftarrow$ **true**
25:     **end if**
26: **end while**
27: **return** $Y$

---

## C.2 Case Study

We provide detailed cases of reasoning process of T2M on the three benchmarks.

> Case on **HiToM**
> **Story:** "The following story happens in chronological order. You will be given a multiple-choice question and a note at the end. First give step-by-step analysis about the question. Then output the answer.
> 1 Sophia, Jacob, Lily, Lucas and Alexander entered the bathroom.
> 2 The green_pepper is in the blue_suitcase.
> 3 Sophia moved the green_pepper to the blue_bottle.
> 4 Sophia exited the bathroom.
> 5 Jacob made no movements and stayed in the bathroom for 1 minute.
> 6 Jacob exited the bathroom.
> 7 Lily moved the green_pepper to the green_pantry.
> 8 Lily exited the bathroom.
> 9 Lucas made no movements and stayed in the bathroom for 1 minute.
> 10 Lucas exited the bathroom.
> 11 Alexander moved the green_pepper to the blue_suitcase.
> 12 Alexander exited the bathroom.
> 13 Sophia, Jacob, Lily, Lucas and Alexander entered the waiting_room."
>
> **Question:** "Where is the green_pepper really?"
>
> **Choices:** "A. blue_suitcase, B. red_bottle, C. green_pantry, D. green_envelope, E. blue_bottle, F. red_cupboard, G. green_cupboard, H. green_bathtub, I. green_basket, J. green_crate, K.

blue_crate, L. blue_bathtub, M. blue_cupboard, N. red_suitcase, O. red_container"

**Response:**
`<think>`
Okay, let's try to figure out where the green_pepper is. The story is in chronological order, so I need to follow each step carefully.

First, the initial state: The green_pepper is in the blue_suitcase. That's step 2. Then, step 3 says Sophia moves it to the blue_bottle. So after that, the green_pepper is in the blue_bottle. Then Sophia exits (step 4).

Next, Jacob stays for a minute (step 5) and exits (step 6). No movement here, so the green_pepper remains in the blue_bottle. Then Lily moves it to the green_pantry (step 7), so now it's in the green_pantry. She exits (step 8).

Then Lucas stays for a minute (step 9) and exits (step 10). No movement, so the green_pepper is still in the green_pantry. Then Alexander moves it to the blue_suitcase (step 11), so now it's back in the blue_suitcase. He exits (step 12).

Finally, everyone enters the waiting_room (step 13). The question is asking where the green_pepper really is. The last action was Alexander moving it to the blue_suitcase. So the final location should be the blue_suitcase. Let me check again to make sure I didn't miss any steps.

Yes, step 11 is the last movement. So the answer should be blue_suitcase.

Now I need to choose an answer based on my intuition from: A. blue_suitcase, B. red_bottle, C. green_pantry, D. green_envelope, E. blue_bottle, F. red_cupboard, G. green_cupboard, H. green_bathtub, I. green_basket, J. green_crate, K. blue_crate, L. blue_bathtub, M. blue_cupboard, N. red_suitcase, O. red_container
`</think>`

**Answer:** A. blue_suitcase

Case on **ToMATO**
**Conversation:** Liam Johnson: "Hi!"
Ethan Johnson: "Hi, how are you?"
Liam Johnson: "Ethan, I've noticed you've been a bit distant lately, is everything okay?"
Ethan Johnson: "Yeah, everything's fine, just been really busy with the restaurant and some personal stuff, you know how it is."
Liam Johnson: "Come on, Ethan, we've been friends for years, I can tell when something's bothering you. What's really going on?"
Ethan Johnson: "Alright, alright... to be honest, I've been having some trouble making ends meet, the restaurant's been slow and I'm falling behind on some bills, but I'll figure it out, I always do."
Liam Johnson: "Ethan, you're not going to 'figure it out' on your own, you need a plan and some guidance. Let me take a look at your finances and we can go over some options together, okay?"
Ethan Johnson: "Man, I appreciate that, Liam, but I don't want to be a burden, I can handle it, I just need to get my head out of my... okay, fine, yeah, maybe a fresh set of eyes would help, thanks."
Liam Johnson: "Alright, let's get down to business then. I'll take a look at your books and we'll get a plan in place. But next time, don't be so stubborn and come to me sooner, got it?"
Ethan Johnson: "Ha! Yeah, yeah, got it, I'll try to work on that, thanks, Liam. And thanks again, man, I owe you one... or a few thousand, it seems."
Liam Johnson: "Yeah, yeah, just pay me back in free meals at the restaurant, and we'll call it square. But seriously, let's take a close look at your expenses and see where we can cut back,

okay?"

Ethan Johnson: "Aww, man, you're a lifesaver, Liam. Yeah, let's dig in and see where I can tighten the belt, I'm sure there's some fat to trim. And don't worry, you'll be eating like a king at my place for a long time to come."

Liam Johnson: "Alright, let's get to work then. I'll start going over the numbers and we can discuss some options. And don't make promises you can't keep, just focus on getting back on your feet, okay?"

Ethan Johnson: "Yeah, yeah, got it, thanks for keeping me grounded, Liam. I'll let you do your magic with the numbers, and I'll focus on getting the restaurant back on track. You're right, one step at a time."

Liam Johnson: "That's the spirit, Ethan. Just remember, I'm not a miracle worker, you need to be effort too. Let's schedule a follow-up meeting for next week and we'll reassess where we're at, sound good?"

Ethan Johnson: "Absolutely, sounds like a plan, Liam. I'll make sure to have all my ducks in a row by then. And thanks again, man, just knowing you've got my back makes me feel a hell of a lot better."

**a0:** "He feels a sense of dry amusement at Ethan's joke, but also a bit of unease about the seriousness of his financial situation"

**a1:** "He feels a bit frustrated that Ethan is not being open about his struggles, and he's worried that he might be hiding something"

**a2:** "He feels a bit anxious about bringing up the sensitive topic of Ethan's financial struggles, but he want to help his friend"

**a3:** "He feels a sense of relief that Ethan is finally opening up and accepting help, but also a bit of irritation that he didn't listen to him sooner"

**q:** "How does Liam feel when he says "Come on, Ethan, we've been friends for years, I can tell when something's bothering you. What's really going on?""

**response:**
<think>
Okay, let's see. The question is about how Liam feels when he says that specific line. Let me look at the conversation again.

So, Liam starts by greeting Ethan. Then Ethan says he's been busy with the restaurant and personal stuff. Liam responds with "Come on, Ethan... What's really going on?" That line seems like Liam is pushing Ethan to be honest.

Earlier, Ethan had said he's been busy, but Liam is pointing out that they've been friends for years and he can tell something's up. So Liam is probably feeling concerned or worried about Ethan. He's trying to get Ethan to open up because he's noticed something's wrong.

Looking at the options, even though the options aren't listed here, the possible answers would likely be emotions like concern, frustration, patience, or maybe anger. But based on the context, Liam is a friend trying to help, so it's more about concern and wanting to support Ethan. He's not angry, but rather trying to get Ethan to talk. So the answer is probably that Liam feels concerned or worried.

Now I need to choose an answer based on my intuition from: [A] He feels a sense of dry amusement at Ethan's joke, but also a bit of unease about the seriousness of his financial situation

[B] He feels a bit frustrated that Ethan is not being open about his struggles, and he's worried that he might be hiding something

[C] He feels a bit anxious about bringing up the sensitive topic of Ethan's financial struggles, but he want to help his friend

[D] He feels a sense of relief that Ethan is finally opening up and accepting help, but also a bit

of irritation that he didn't listen to him sooner
```
</think>
```

**Answer:** B

---

Case on **ToMBench**
**STORY:** "Zhang Wei is the performance champion of the department, but he encounters some challenges in his recent project, which causes his performance to be not good. Li Yun is the department manager, he always supports Zhang Wei, hoping to fight for the best evaluation for him in the year-end review. Chen Tao is Zhang Wei's colleague, he also hopes to get promoted in the year-end review. This afternoon, Li Yun quickly browses Zhang Wei's review form in the office. When Zhang Wei passes by Li Yun's office, Li Yun looks up at him and nods to him. Zhang Wei gives a slight smile to Li Yun and continues to walk past. Chen Tao sees all this in the tea room not far away."

**QUESTION:** "Why does Li Yun nod to Zhang Wei?"

**OPTION-A:** "Li Yun just sees Zhang Wei's review form and thinks his performance is good."
**OPTION-B:** "Li Yun wants to tell Zhang Wei that he gets a high evaluation."
**OPTION-C:** "Li Yun is greeting Zhang Wei."
**OPTION-D:** "Li Yun nods to Zhang Wei unintentionally."

**ANSWER:** "B"

**map:** {"A": "D", "B": "C", "C": "A", "D": "B"}

**output:**
```
<think>
```
Okay, let's try to figure out why Li Yun nodded to Zhang Wei. So, the story says that Zhang Wei is the performance champion of the department, but he's having some challenges in his recent project, which is affecting his performance. Li Yun is the department manager who supports Zhang Wei and wants to fight for the best evaluation for him in the year-end review. Chen Tao is another colleague who also wants to get promoted.

Now, the key moment is when Li Yun quickly browses Zhang Wei's review form in the office. Then, when Zhang Wei passes by Li Yun's office, Li Yun nods at him, and Zhang Wei gives a slight smile and continues walking. Chen Tao sees this from the tea room.

So, why would Li Yun nod to Zhang Wei? Let's think. Since Li Yun is the manager and wants to help Zhang Wei get the best evaluation, maybe he's trying to show support. But why nod specifically? Maybe because he just looked at Zhang Wei's review form. If he's looking at the form, perhaps he's checking Zhang Wei's performance or the evaluation. But since Zhang Wei is having challenges, maybe Li Yun is trying to encourage him. Alternatively, maybe Li Yun is trying to signal that he's aware of Zhang Wei's situation and is supporting him.

Alternatively, could it be that Li Yun is trying to get Zhang Wei to know that he's being considered for a good evaluation? Or maybe Li Yun is trying to make sure that Zhang Wei's performance isn't overlooked despite the recent challenges. The nod could be a gesture of support or a reminder that Li Yun is on his side.

Another angle: since Chen Tao is also hoping for promotion, maybe there's some competition. But the question is about Li Yun's action towards Zhang Wei. The story says Li Yun always supports Zhang Wei and wants to fight for the best evaluation. So the nod is likely a supportive gesture.

But why would Li Yun look at the review form before nodding? Maybe he's checking the form and wants to confirm that Zhang Wei's performance is being considered, or

maybe he's trying to influence the evaluation. The nod could be a way to communicate that he's aware of Zhang Wei's situation and is advocating for him.

Alternatively, maybe Li Yun is trying to get Zhang Wei to acknowledge his support, but the story says Zhang Wei just gives a slight smile and continues. So the nod is probably a sign of support or encouragement.

Looking at the answer options, even though they aren't listed here, the options would likely be something like: A. To show support, B. To check his performance, C. To compete with Chen Tao, D. To inform about the review. But since the options aren't given, based on the story, the most logical answer is that Li Yun is supporting Zhang Wei, so the answer would be something like A. But since the user hasn't provided the options, I have to assume that the answer is based on the given context. The answer is likely related to support or advocacy for Zhang Wei's evaluation. So the answer index would be the one that matches that reasoning.

Now I need to choose an answer based on my intuition from: A. Li Yun nods to Zhang Wei unintentionally. B. Li Yun is greeting Zhang Wei. C. Li Yun just sees Zhang Wei's review form and thinks his performance is good. D. Li Yun wants to tell Zhang Wei that he gets a high evaluation.
```
</think>
```

**Final Answer:** D

