# OpenReview forum: "To Think or Not To Think, That is The Question for LLM Reasoning in Theory of Mind Tasks"
_ICLR.cc/2026/Conference — Submitted to ICLR 2026_

### Official Review · Reviewer_3FPz · 2025-10-24

**Soundness:** 1
**Presentation:** 1
**Contribution:** 1
**Rating:** 2
**Confidence:** 1

**Summary:**

n/a - please see ethics review. I believe this paper should be desk rejected because of its field.

**Strengths:**

n/a.

**Weaknesses:**

n/a.

**Questions:**

n/a.

**Details Of Ethics Concerns:**

I no longer think it is ethical to work on Theory of Mind in academia, and I think we should be actively discouraging development of these methods. I know that's a strong statement; please bear with me for a minute. I wrote one of the papers referenced heavily in this work and others like it, and I have since found out that my work has since been used for unethical applications including persuasion, marketing, and political advertising, in academia and in industry. Yes there are upsides to Theory of Mind – that's why I and many others started working on it in the first place. Better affective technologies, agents that can anticipate your needs without you asking, therapeutic agents, education agents...etc. But I think while these are indeed upsides, we can achieve these goals in other ways that leave the user's agency and mind uninvaded by AI's – e.g., by creating tools that users can have an easier time adapting to and learning, rather than tools that anticipate and adapt to the user. Essentially, I think the upsides are limited and clever solutions could get to the same ends in less invasive ways. And the downsides are basically infinite. If we're successful in building agents that really can understand what we're thinking and feeling without saying it, those agents will be used by big companies and people in power to successfully simulate our thinking and manipulate us into doing what they want. This isn't conjecture or conspiracy theory; each major tech company is actively investigating AI persuasion and marketing for exactly this reason, as are political campaigns and governments throughout the world. I would know, because many of them have reached out to me to help them implement their methods. As people, we should be scared by this.

There's a counter argument of course, which is that this is a cat and mouse game like security, and that academia shouldn't attempt to restrict research into these methods: it's best to have this all out in the open so that people can research defenses. While it's possible that defenses could be created against this type of technology, I think that is a dangerous and unproven hypothesis to rely on. It may be the case that this is a setting in which offense is much easier than defense, especially because unlike in computer security, the defender is ultimately a human, not an algorithm. The best and most secure defense would be to train humans to think critically about these types of attacks, and that is a very hard thing to do, especially as the attacks get more and more sophisticated and human education systems become increasingly politicized and unequal. I think it is much better to keep this pandora's box closed for as long as possible, and to work on functional uses of AI that have fewer ethical downsides. I would encourage the ethics committee to think carefully about this, discuss among themselves (and maybe reach out to me if you want to chat more), and do as much as possible to prevent works like this from being published in top conferences (which increases the visibility and incentive to work on this subject).

---

> ### Author Response · Authors · 2025-11-24
>
> We are very grateful to the reviewer for raising this profound and good-faith ethical concern. We fully understand and share your concern for protecting human agency and preventing the malicious use of AI. The gravity of this issue, especially given the reviewer's professional perspective, resonates deeply with us.
> However, we would like to respectfully argue that we (the academic community) must deeply understand the ToM capabilities of large language models, precisely in order to prevent the negative impacts they might have.
>
> ## 1. Technology is a Double-Edged Sword; Evading Research Cannot Solve the Root Problem
> Technology itself is neutral. As the reviewer knows, it is not just ToM; the entire history of AI development has been accompanied by deep concerns about its safety. But the immense contributions of AI to science, medicine, and education have led us to choose to vigorously develop safety and alignment techniques alongside active research, rather than halting research.
> The same applies to ToM. As the reviewer noted, this technology is already being actively researched. If the academic community stops exploring ToM, those actors who intend to misuse the technology will not stop. This would not only fail to solve the security problem at its root but would, in fact, leave the defenders in an "intelligence deficit," making us incapable of building effective guardrails.
>
> ## 2. Our Work is a Critical Diagnosis, Not a Capability Advancement
> Critically, our work is a critical diagnosis of current capabilities, not a capability advancement. Our research reveals the uniqueness of the ToM task and discovers that even advanced LRMs have significant problems on this task .
> Specifically, we diagnosed failure modes such as "slow thinking collapse" (where accuracy drops as responses grow longer) and the "option matching shortcut" (where models rely on superficial matching rather than genuine deduction) . We believe this clear diagnosis of current models' limitations provides a necessary empirical foundation for future evaluation and research into ToM, including its potential safety implications.
>
> ## 3. Ethical Statement
> We fully agree with the ethical sensitivity of this research. Based on your invaluable feedback, we have add a dedicated "Ethical Statement" section to the revised version. In this section, we will discuss the dual-use nature of ToM technology in detail, address the concerns you have raised, and explicitly state that our work aims to contribute to the critical understanding and safe evaluation of the field, rather than blindly advancing its capabilities.

---

> > ### Comment · Reviewer_3FPz · 2025-11-25
> > **Response**
> >
> > Thank you for these thoughts! I respectfully disagree with them.
> >
> > I don't think technology itself is neutral in all respects. Some technologies are more dangerous than others. Bioweapons, for example, are the exact kind of research topic that has a "pandora's box" quality. Working on bioweapons might have some upsides, but it has almost infinitely many downsides, especially if that research is done out in the open where bad actors can take advantage of it. I'm sure people are working on this, but I for one am grateful that this isn't a heavily researched area, and most of this research is secret so various underresourced autocrats aren't capable of accessing it. Why is ToM so different from bioweapons? Or more specifically - how do you know the downsides of ToM aren't similarly scary? You'd have to assume a lot about how good ToM could become to answer that, and if you're wrong the downsides are enormous.
> >
> > Second - critical diagnoses are essential to advancing capabilities. That's the basic argument you're making for why your paper is relevant to ToM, right? I don't think you can have that both ways, unfortunately.
> >
> > Thanks for the thoughtful response, but I won't be changing my score. I think the AC's should decide whether my review should be considered for this paper, given my perspective on the ethics of the research area.

---

### Official Review · Reviewer_VWm9 · 2025-10-31

**Soundness:** 2
**Presentation:** 2
**Contribution:** 2
**Rating:** 4
**Confidence:** 3

**Summary:**

The paper performed systematic analysis contrasting the theory-of-mind performance of reasnoning and non-reasoning models. The results show the “resoning” abilities that advance the performance on analytical tasks does not consistently bring improvement on ToM tasks.

**Strengths:**

The paper provides a quantitative study on the behaviour of reasoning models, covering several models and datasets, revealing several behavioral patterns of the reasoning models on this type of task.

**Weaknesses:**

1. In general, the studies in the paper provide "correlation" between different behaviours, e.g. long response, i.e. extended thinking, is correlated with lower performance. The reviewer thinks this does not suffice for the causality conclusions made in the paper, e.g. slow thinking leads to reasoning collapse.
2. The methods proposed by the paper (stop the thinking process when "wait" appears too many times; remove options in the prompt to force "thinking") are straightforward; the technical contribution is limited.

**Questions:**

1. line 204
> "reasoning-tuned models can be more prone to error when confronted with deeply nested inference"

The performance of gpt series int Table 2 does not support this claim, as the reasoning variants (o3 and o4mini) are consistently better than their non-reasoning variant (4o).

2. In Table 1 and Table 2, the results are mixed, with no clear pattern across model families or datasets. E.g. the conclusion in Section 4.1.1, "reasoning collapses in higher-order inference" is not supported by gpt series and Qwen3-32B series. The discrepancy of behaviours in different models is not discussed.

3. Pattern in Figure 1 also exhibits in analytical tasks. The reason behind this pattern in those tasks can be that when a problem is too difficult, the model tends to repeatedly generate useless reasoning steps. Simply from Figure 1 it's hard to draw the conclusion that the excessive reasoning is the **cause** of the failure. It could be that when a problem is out-of-distribution, and the model simply cannot solve it, its failure will be associated with a long generation. The correlation between performance and response length cannot contribute to conclusions about the causal effect of extended reasoning on performance, i.e. the first conclusion in Section 4.3.1 is not supported.

4. In Section 4.3.2, for the comparison between ToM and "formal reasoning", there is no experiments on formal reasoning in the paper. The comparison is unclear and less supported. It would be helpful to report concrete numbers on different domains and then compare them.

5. How does the proposed S2F method differ from [1]?

[1] Zhang, Junyu, et al. "AlphaOne: Reasoning Models Thinking Slow and Fast at Test Time." arXiv preprint arXiv:2505.24863 (2025).

---

> ### Author Response · Authors · 2025-11-24
>
> Thank you so much for the detailed comments.
>
> ## For W1
> We have polished the presentation regarding this point in our revised version. For example, we change the statement of the first conclusion to "Slow Thinking Correlates with Reasoning Failures." Besides, we include a new experiment in Sec. 4.2.4 to support the insight.
>
> ## For W2
> The main objective of this paper is to identify the problems in applying reasoning to ToM. Therefore, our core contribution is the dicussion and analysis of the insights behind reasoning failure in ToM. We develop the methods to further verify the problems we identified and to point out potential research avenues. They are preliminary exploration inspired the reasoning issues, offering valuable support for our conclusions. To clarify the methods are not dedicated contributions, we move it to Sec. 4 as part of our analysis.
>
> ## For Q1
> We have polished this sentence in our revised version to “some reasoning models are susceptible to reasoning collapse when confronted with complex scenarios.”
>
> ## For Q2
> We have updated the title of Sec 4.1.1 to “Reasoning Loses Dominance in High-Order Inference”. Besides, we have polished the discussion in this section to ensure preciseness. For example, we interpret the results as “some reasoning models are susceptible to reasoning collapse when confronted with complex scenarios”.
>
> ## For Q3
> We have revised our first conclusion in Sec. 4.2.1 to 'Slow Thinking Correlates with Reasoning Failures.' This statement is supported by two key lines of evidence: First, Sec. 4.2.1 and 4.2.2 establish the correlation between response length (reasoning effort) and performance decline. Second, our experiments in Sec. 4.2.4 and 4.3.1 demonstrate that constraining reasoning length yields clear improvements compared to unconstrained reasoning.
>
> ## For Q4
> We observe the difference between ToM and “formal reasoning” is two-fold. 1) According to [1,2], for example, DeepSeek-R1 clearly outperforms DeepSeek-V3 on most benchmarks, a trend that is also observed with Qwen3-8B-reasoning surpassing Qwen3-8B. Conversely, DeepSeek-R1 lags behind DeepSeek-V3 on HiToM and ToMATO. Likewise, Qwen3-8B-reasoning exhibits inferior performance compared to Qwen3-8B on these tasks. 2) According to [3], options on formal reasoning such as math significantly improve performance, but have a bad effect on ToM reasoning in Sec. 4.2.5.
>
> ## For Q5
> AlphaOne dynamically controls the reasoning condition. Firstly encourage slow thinking by inputting “wait” and finally switch to fast thinking by inputting </think>. However, S2F monitors “wait” rather than actively input it. Because the goal is to stop slow thinking on complex scenarios, the input of “wait” is useless. We only intervent the reasoning process when it is too long.
>
> [1] Guo, Daya, et al. "Deepseek-r1 incentivizes reasoning in llms through reinforcement learning." Nature 645.8081 (2025): 633-638.
>
> [2] Yang, An, et al. "Qwen3 technical report." arXiv preprint arXiv:2505.09388 (2025).
>
> [3] Raman, Narun, Taylor Lundy, and Kevin Leyton-Brown. "Reasoning models are test exploiters: Rethinking multiple-choice." arXiv preprint arXiv:2507.15337 (2025).

---

### Official Review · Reviewer_iDRt · 2025-10-31

**Soundness:** 3
**Presentation:** 3
**Contribution:** 2
**Rating:** 6
**Confidence:** 5

**Summary:**

The paper “To Think or Not To Think” investigates whether reasoning-oriented language models genuinely improve Theory of Mind (ToM) capabilities. Evaluating 11 models across three ToM benchmarks, the authors find that reasoning-tuned models often perform no better—and sometimes worse—than their non-reasoning counterparts. Through detailed behavioral analysis, they identify two key issues: slow-thinking collapse, where longer reasoning chains degrade accuracy, and option-matching shortcuts, where models rely on surface cues instead of true understanding. To address these, they propose two strategies—S2F, which trims unproductive reasoning, and T2M, which separates reasoning from answer selection—both improving ToM performance and highlighting fundamental differences between formal reasoning and social cognition in LMs.

**Strengths:**

- The paper evaluated a fair number (11) of reasoning models from different model families, which makes the analysis more compelling.
- The paper conducted a fairly comprehensive analysis of when and why the reasoning model fails to outperform, which provides valuable insights to the LLM ToM community.
- Their proposed interventions, S2F (“stop slow-thinking failure”) and T2M (“think-then-match”), offer measurable gains.

**Weaknesses:**

- The "causality vs. correlation" problem: the paper's claim “reasoning tuning hurts ToM” is supported empirically, but confounds (training data, safety filters, decoding defaults) aren’t fully isolated. The paper compares across different model families, not controlled ablations.
- Claims about “formal vs social reasoning” rest only on the evaluation of three datasets. This small number of datasets makes the conclusion less convincing.
- Most of the analysis is done only on a single benchmark (Hi-ToM).
- The proposed methods constitute a main contribution of the paper. Yet they are pretty preliminary, and the paper lacks enough analysis of the methods. Also, the S2F method has minimal impact on ToMBench (and mostly on ToMATO).

**Questions:**

- Does "vanilla" and "CoT" prompting for reasoning models mean disabling and enabling thinking mode?

---

> ### Author Response · Authors · 2025-11-24
>
> Thank you so much for acknowledging the compelling analysis, valuable insights, and measurable gains of our paper
>
> ## For W1
> Thanks for your review. We have revised our presentation regarding this point. Our experiments contain both controlled reasoning and non-reasoning groups and additional reasoning models. The main goal is to investigate the overall effectiveness between LRMs and LLMs. It is hard to isolate all the potential variables. Therefore, our experiments aim to provide as much as possible insights from both macro and micro perspective through the comparison of different models under the same hyperparameter setting.
>
> ## For W2
> Our selection of benchmarks cover relatively comprehensive scenarios in ToM, spanning from thinking order and taxonomy. We believe these benchmarks are both widely-used and representative.
>
> ## For W3
> Most of the experiments are conducted on all the three benchmarks. Because Hi-ToM is the most challenging one, our analysis is mainly based on the results of Hi-ToM. However, we have provided the additional experimental results in Appendix. In our revision, we also include results from ToMATO and ToMBench in the main paper, and we discuss the results more rigorously.
>
> ## For W4
> The main purpose of introducing this method is to further verify the problems we identified and to point out potential research avenues. Although these results are preliminary, they offer valuable support for our conclusions. Besides, we have polished the discussion in the revised manuscript regarding S2F's performance. The reason S2F shows limited improvement on ToMBench is that “As discussed in Sec. 4.2.1, the detrimental effects of slow thinking are less pronounced on simpler benchmarks, such as ToMATO. Consequently, the benefits gained from inhibiting slow thinking are counterbalanced by the disruption to the natural reasoning process, resulting in negligible overall improvement or even a slight decline in performance.”
>
> ## For Q1
> "Vanilla" for reasoning models means the raw thinking mode. "CoT" means CoT prompt is aditionally added.

---

### Official Review · Reviewer_Qmcm · 2025-11-01

**Soundness:** 2
**Presentation:** 2
**Contribution:** 2
**Rating:** 4
**Confidence:** 4

**Summary:**

Based on the discussion around the reasoning advances of large reasoning models and their benefits in various applications compared to other types of non-reasoning large language models, this paper discusses the nuances of comparing reasoning and non-reasoning models in Theory of Mind (ToM) as one of the interesting applications of large language models. ToM involves unique characteristics not found in other types of reasoning benchmarks, such as math or logic. The authors find that reasoning in large reasoning models can be detrimental to ToM tasks, where specifically long chains of reasoning collapse the reasoning abilities of the models. They further find that there are certain shortcuts in benchmarks that models tend to utilize, and these shortcuts, in fact, make their reasoning abilities fragile and prevent them from true, unbiased reasoning.

**Strengths:**

The paper is interesting in terms of the problem it tackles, examining the advantages of reasoning in large reasoning models for ToM-type tasks that are inherently different from other types of reasoning, such as mathematical reasoning. Their findings also expand upon other results in the literature on overthinking and efficient reasoning, where similar problems with lengthy reasoning have been discussed.

The paper is also comprehensive in terms of the experimental setup, covering a sufficient number of benchmarks and models from different families to make the findings more robust. It is relatively well-written and easy to follow in terms of the flow of arguments and reasoning provided by the authors.

There are also interesting experiments with insightful results, for instance:
– The moderate thinking analysis, where they compare reasoning models with non-reasoning models that are prompted to perform chain-of-thought reasoning;
– The evaluation of reasoning effort in GPT models; and
– The finding that harder questions are sometimes complementarily solved by non-reasoning and reasoning models.

**Weaknesses:**

Some of the comparisons in the paper are questionable, which in turn makes some statements and arguments less clear and less defensible:

1.	You argue that reasoning models think longer, and this is evident in the cases where they fail in reasoning (Figure 1). But why have you compared the R1 model with the Claude model here? Wouldn't a better comparison be between R1 and V3, which belong to the same family?

2.	Related to the same topic, apart from the figure that has the mentioned issue, you have not discussed anywhere else what the length distribution looks like for non-reasoning models. When questions are more difficult, you show that reasoning models produce much longer responses with incorrect reasoning. This, to some extent, is attributed to their characteristic verbosity, but it could also be because of the increased difficulty of the problem, which causes them to reason more. This might be evident in non-reasoning models as well. I actually think the authors' argument is intuitive and probably correct, but the evidence in the paper does not sufficiently support it.

3.	The authors have included discussions of the performance of the Grok and Claude models, listed as reasoning models, and have also compared their performance with R1, for instance. I’m not sure if I follow why these two models are included, especially when they are compared with other models not in their family and thus not directly comparable. This makes the arguments about them less informative and, to some extent, flawed.

4.	Why have you conducted the S2F experiments only with the Qwen models and not with others? I don’t want to be pedantic and ask for every experiment to be run on all models, but the current setup feels somewhat arbitrary. The same argument holds for the T2M experiments.

Some details in the paper are inconsistent or missing, which makes replicability difficult. For instance, the paper hasn't mentioned any evaluation metric or evaluation criteria used, or if it has, the explanation is unclear. Other issues include inconsistent use of model names, for example, sometimes you mention Qwen-32B-Reasoning, and in other places R1-Distill-Qwen-32B.

Since the authors mention System 1 and System 2 as a potential approach to finding a good dynamic system that combines the abilities of reasoning and non-reasoning models, it would be good to cross-check and reference the following paper as well:

Ziabari, A. S., Ghazizadeh, N., Sourati, Z., Karimi-Malekabadi, F., Piray, P., & Dehghani, M. (2025). Reasoning on a spectrum: Aligning LLMs to System 1 and System 2 thinking. arXiv preprint arXiv:2502.12470.

Similarly, the authors haven’t connected their arguments with findings in the overthinking literature, such as the following work. Adding this would enhance both clarity and the connection between their work and the broader line of research on this subject:

Sui, Y., Chuang, Y. N., Wang, G., Zhang, J., Zhang, T., Yuan, J., … & Hu, X. (2025). Stop overthinking: A survey on efficient reasoning for large language models. arXiv preprint arXiv:2503.16419.

**Questions:**

Please respond to the issues that I have raised above.

---

> ### Author Response · Authors · 2025-11-24
>
> Thank you so much for acknowledging the interestingness, comprehensiveness and insightful results of our paper.
>
> ## For W1
> The comparison of R1 and Claude aims to demonstrate why Claude achieves the SOTA while R1 fails to achieve a great performance. Figure 1 shows R1 takes prolonged thinking, making the incorrect answer. The reason for not comparing R1 and V3 in this figure is that there is no default thinking process in the output of V3. Instead, we compare response length between V3-CoT and R1 in Figure 3. V3-CoT significantly outperforms R1, with a response length that aligns with Claude's. This observation highlights that moderate thinking is sufficient for superior performance.
>
> ## For W2
> Since the benchmark prompts require LLMs to answer directly, the response only consists of the answer letter. Therefore, regardless of the complexity of the tasks, there is no increasing response length in the non-reasoning models. Moreover, we provide the response length distribution of V3-CoT in Figure 4, showing the response length of non-reasoning models on both correct and incorrect answers tends to be short.
>
> ## For W3
> Our main goal is to compare LRMs and LLMs to investigate the effectiveness of the reasoning ability. The comparison is two-fold. First, from a macro perspective, we can observe the overall performance difference between reasoning and non-reasoning models. Second, we also designed rigorous control studies to understand the findings from the micro level. The introduction of Grok and Claude aims to derive more insights from macro perspective. Because they are trained by different strategies, involving different reasoning models can help us draw a comprehensive conclusion on their patterns. For example, from the comparison of R1 and Claude in Figure 1, we know the response length is a signature of failure. To help our readers understand the functions of two-fold comparison, we illustrate our study design in Sec. 3.1.
>
> ## For W4
> Since S2F and T2M require real-time monitoring and intervention, we can only implement it on open-source models like Qwen3. Therefore, apart from Qwen3 series, we additionally involve R1-Distill-Qwen series to comprehensively evaluate. To provide more information on the experimental results, we further conduct experiments on R1-Distill-Llama-8B, as shown in Sec. 4.3.1. The performance on HiToM increases by 13.9%, while there is a slight decrease on ToMATO and ToMBench. With the newly added model, our experiment covered the most representative open-source model families, justifying our identified problem “Slow Thinking Correlates with Reasoning Failures”.
>
> ## For Other Issues
> We have clarified the evaluation metric in Sec. 3.1 of our revised version “All the experimental results are evaluated by the accuracy.” The mentioned references have been also added in our revised version.

---

> > ### Comment · Reviewer_Qmcm · 2025-11-24
> >
> > I thank the authors for their reply.
> >
> > Based on your further explanation, the reason behind the experimental setup is more clear now. However, since they are still far from a controlled experiment and many things are different between the models that are tested here, the findings and takeaways will be more precise if they avoid the causal language used in the paper.

---

> > > ### Author Response · Authors · 2025-11-29
> > >
> > > Thank you so much for raising the score to 6. We have carefully polished our presentation in the revised verison. Your feedback has been invaluable in enhancing the quality of our work, and we remain committed to further polishing the final submission.

---

### Official Review · Reviewer_auky · 2025-11-01

**Soundness:** 2
**Presentation:** 4
**Contribution:** 2
**Rating:** 4
**Confidence:** 4

**Summary:**

This paper investigates whether the reasoning abilities of Large Reasoning Models, which have achieved major success in structured domains like mathematics and code, also have improvements in Theory of Mind tasks. The author is asking: Does explicit reasoning actually improve ToM performance, or it actually worsen the performance, and why.

The author used 11 LLMs (7 reasoning, 4 non-reasoning) and test their performance on 3 ToM benchmarks. The main finding is: reasoning models fail to outperform non-reasoning counterparts across most benchmarks. In specific, in high-order inference, reasoning model performs badly.

The author analyzed the reason: slow thinking on complex ToM tasks leads to failure (although moderate thinking with CoT prompt on non-reasoning model increases performance); reasoning models rely on matching multiple-choice options rather than genuine deduction and that hurts the performance. These are the difference between formal reasoning and ToM reasoning.

The author gives 2 ways to increase reasoning model's performance on ToM tasks (directly related to the 2 reason why reasoning model performs badly): i) slow-to-fast reasoning: Essentially force the model to end the thinking process when the reasoning is too long (signaled by the multiple usage of "wait"); and ii) Think-to-Match: Forces the model to reason without seeing options first, then introduces options only after generating internal reasoning

**Strengths:**

The paper’s central strength lies in its conceptual reframing of reasoning in large language models from formal to social cognition, challenging the implicit assumption that methods improving logical or mathematical reasoning automatically enhance Theory of Mind.

The paper also discovered that thinking long may not be better in the ToM tasks.

The above summary part has already covered the details.

**Weaknesses:**

### Quality of the Control Study

The control design (section 3.1) relies on comparing different “reasoning” and “non-reasoning” model variants (e.g., Qwen3-Reasoning vs. Qwen3), but these pairs are still different models, differ in multiple confounding factors such as training dataset and reinforcement objectives (you are unlikely to know these training details, to begin with), making it difficult to attribute performance differences purely to the presense/absense of reasoning behavior.

In short, **this is not a good control study** and in my opinion there is a clearly better way to do that: use a within-model manipulation, prompting the same reasoning-tuned model to skip its reasoning process (inject a neutral “answer directly” context template, or fill in some dummy reasoning like "I must answer the question now"), to isolate the causal effect of explicit reasoning itself.

### Justification of Section 4.2.1

One of the paper's main claim is: "slow thinking on complex ToM tasks **leads** to failure". This suggests a causal relationship. However in this section I only see correlation. One could easily argue that if a question itself is difficult, it causes both long reasoning and low accuracy, not that long reasoning caused low accuracy. (more suggestions are in the "Question" section below)

### Other Minor Issues

- Page 2, line 073: "our **ablation study** on HiToM demnstrates that ...": "ablation study" means removing a component of an ML system (for example, an attention head) and then analyzing the resulting performance of the system. So please use another term to avoid confusion;
- Page 3, sec 3.1, first line: There are 11 models in total (7 reasoning+4 non-reasoning). The paper said 10;
- Page 3, sec 3.2, fourth line: "counterintuitive" is a word, please use this instead of "conter-intuitive"

**Questions:**

Please address/justify the weaknesses mentioned in the "weakness" section. Below are my specific questions/suggestions:

### Quality of the Control Study

I would suggest that the author either
- Do the control study mentioned in the "weakness" section, or
- Justify that the original paper's design is enough to attribute performance differences purely to the presense/absense of reasoning behavior, and why my proposal is not better

Also, Claude and Grok models are included in these 11 models but I do not see a reason of doing so, as no meaningful comparison group is set up.

### Justification of Section 4.2.1

There are some causality-related experiments mentioned in the following chapters. In specific, sec 4.4.1 (slow to fast reasoning) can be reframed as a causal evidence. I would suggest that author
- Expand sec 4.4.1 to all models (currently only Qwen is covered), and
- Put them as direct evidence to support your argument that "slow thinking on complex ToM tasks leads to failure", rather than a discussion/exploration

### Other Minor Issues

Please address the minor issues mentioned in the "weakness" section as well

### Citation Cleaning

When a paper appears on both arxiv and a published proceeding/conference, it would be better to mention the conference instead of leaving only an arxiv link. Below are some examples and please self-check the rest

- Yinghui He, Yufan Wu, Yilin Jia, Rada Mihalcea, Yulong Chen, and Naihao Deng. Hi-tom: A benchmark for evaluating higher-order theory of mind reasoning in large language models. arXiv preprint arXiv:2310.16755, 2023. **This paper appears in EMNLP findings, 2023. "T" and "M" should be capitalized**

- Simeng Han, Hailey Schoelkopf, Yilun Zhao, Zhenting Qi, Martin Riddell, Wenfei Zhou, James Coady, David Peng, Yujie Qiao, Luke Benson, et al. Folio: Natural language reasoning with first-order logic. arXiv preprint arXiv:2209.00840, 2022. **This paper appears in EMNLP 2024**

### Additional Questions on experiment details

The link in the "reproducibility statement" shows that "the requested files are not found". I would like the author to briefly explain, that for each of the 11 models you used, whether you call the model's api or do inference locally. A table showing the computation you used is also welcomed.

---

> ### Author Response · Authors · 2025-11-24
>
> Thank you so much for raising the concerns and constructive suggestions.
>
> ## For W1
> Our main goal is to compare LRMs and LLMs to investigate the effectiveness of the reasoning ability. The comparison is two-fold. First, from a macro perspective, we can observe the overall performance difference between reasoning and non-reasoning models. Through the macro comparison in Sec. 3.2 and Sec. 4.1, we learnt that reasoning models don’t consistently outperform non-reasoning models. Their performance show different patterns on different thinking order and taxonomy. Second, we also designed rigorous control studies to understand the findings from the micro level. For example, in Sec. 4.2.4, we carefully compare Qwen3-8B and Qwen3-8B-reasoning to understand the effectiveness of CoT prompt. The micro-level comparisons do not cover all models at the macro level since many of the reasoning models are closed models, making it impossible to compare with their non-reasoning versions with fully controlled studies. To help our readers understand the functions of two-fold comparison, we illustrate our study design in Sec. 3.1.
>
> ## For W2
> For clarity and rigor, we have revised this statement in the paper to “This suggests reasoning failure correlates with response length, especially on complex tasks.” Besides, to further support our statement, we have added new experiments to demonstrate that moderate thinking can lead to higher ToM performance than excessive or insufficient thinking in Sec 4.2.4. It also echoes our results by introducing CoT to non-reasoning models (see Sec. 4.2.4).
>
> ## For Q1
> We have further clarified the experiment design for two-fold comparison at the macro and the micro levels, as our response to W1. We included Claude and Grok to increase the model coverage in the macro-level comparison, which is further clarified in Sec. 3.1 in the main paper.
>
> ## For Q2
> Since S2F requires real-time monitoring and intervention generation processes, we can only implement it on open-source models. Apart from Qwen3, we also include R1-Distill-Qwen and R1-Distill-Llama to comprehensively study the effectiveness. The three models ensure that we have covered most representative open-source model families, Qwen, Llama, and DeepSeek-based models. The experimental results with R1-Distill-Llama are added to Sec. 4.3.1. With all three models, S2F consistently improves the performance on complex benchmarks such as HiToM. They jointly supported our statement “Slow Thinking Correlates with Reasoning Failures” with another newly added experiment about “Study of token control” in Sec. 4.2.4.
>
> ## For Minor Issues
> We have corrected them in the revised version. The link in the “reproducibility statement” should be accessible, please try it again.

---

> > ### Comment · Reviewer_auky · 2025-11-26
> >
> > I thank the author for the update of the paper and the reply.
> >
> > ## W1+Q1
> > Unfortunately, the author does not resolve my concern.
> >
> > > Our main goal is to compare LRMs and LLMs to investigate the effectiveness of the reasoning ability.
> >
> > You want to investigate the effectiveness of the **reasoning ability**, so you definitely need a well-designed control study that isolate the effect of reasoning as much as possible. This is your center argument. However, simply comparing reasoning/non-reasoning model in the same model family is not a well-designed control study. Training dataset, reinforcement objectives and a lot of things are not controlled, or straight up being unknown. These are 2 different models.
> >
> > And I'm not being delibrately picky or making things difficult for you, because modifying context template to induce (or force) the reasoning model to skip reasoning is not a hard or unreasonable thing to do at all. In fact, the idea is very similar to your 4.3.1 (dynamic intervention). Under this setting, you can just cut the reasoning part completely, without changing the model at all.
> >
> > ##  W2+Q2
> >
> > After this revision, the conclusion of Section 4.2.1 becomes significantly weaker and no longer provides a satisfactory answer to the question of why reasoning fails to outperform. As mentioned before, the most compelling causal evidence comes from Section 4.3.1, yet the paper does not clearly frame its conclusions around this insight.
> >
> > Furthermore, the analyses remain largely behavioral and superficial (e.g., “long thinking hurts performance, moderate thinking helps”), without addressing the underlying question of why these patterns arise. The paper would be substantially stronger if the authors included a deeper discussion of the model’s internal mechanisms.
> >
> > ## Citation Cleaning
> > I ask the author to carefully fix the citations ("please self-check the rest").
> > - Yufan Wu, Yinghui He, Yilin Jia, Rada Mihalcea, Yulong Chen, and Naihao Deng. Hi-tom: A benchmark for evaluating higher-order theory of mind reasoning in large language models. In Findings of the Association for Computational Linguistics: EMNLP 2023, pp. 10691–10706, 2023. **The benchmark name is Hi-ToM, please capitalize "T" and "M". You can use curly bracket in your reference.bib to preserve the capitalized status.**
> > - Chunkit Chan, Cheng Jiayang, Yauwai Yim, Zheye Deng, Wei Fan, Haoran Li, Xin Liu, Hongming
> > Zhang, Weiqi Wang, and Yangqiu Song. Negotiationtom: A benchmark for stress-testing machine
> > theory of mind on negotiation surrounding. arXiv preprint arXiv:2404.13627, 2024. **This paper appears in EMNLP 2024 findings.**
> > - Ruirui Chen, Weifeng Jiang, Chengwei Qin, and Cheston Tan. Theory of mind in large language
> > models: Assessment and enhancement. arXiv preprint arXiv:2505.00026, 2025. **This appears in ACL 2025**
> > - *(inter alia)*
> >
> > ## Experiment Details
> > I would like to repeat the previous review
> > > I would like the author to briefly explain, that for each of the 11 models you used, whether you call the model's api or do inference locally. A table showing the computation you used is also welcomed.
> >
> > ## Conclusion
> > The authors argue that the reasoning abilities of Large Reasoning Models cannot be effectively transferred to improve the socio-cognitive skills required for Theory of Mind, and they provide an analysis of possible causes and solutions. While the topic is interesting, neither the core ideas nor the experimental rigor and reasoning analysis meet the standards of the ICLR main conference. In particular, the primary experiment (4.1) contains methodological issues that should be addressed, and the reasoning analysis remains too superficial, without probing the underlying model mechanisms.
> >
> > As a result, I am maintaining my current rating. I remain open to further in-depth discussion with the authors.

---

> > > ### Author Response · Authors · 2025-12-01
> > >
> > > Thanks for your following response.
> > >
> > > ## For W1+Q1
> > >
> > > We have tested the aforementioned method of prompting to skip reasoning. Beside a few models that will not disclose the raw reasoning process, such as o3, we find that this technique works only on a small part of models in our paper, i.e., Qwen3 series. On more models like DeepSeek-R1, it fails to skip the reasoning process. For example, if we prompt the LLMs to
> > > “Read the following story and answer the question by selecting the correct choice. Output your final answer strictly in the format: Answer: [Letter], DO NOT include any thinking process and directly give the answer.
> > > [Story]....... [Question]...... [Options]......
> > > ”(from HiToM), the results are shown below.
> > >
> > > DeepSeek-R1   “<think>Okay, let's try to figure out where William thinks Emily thinks Owen thinks the pumpkin is. Hmm, this is a bit of a chain of thoughts. Let's break it down step by step…..”
> > >
> > > Claude-Sonnet-4 “I need to trace through the story to understand what William thinks Emily thinks Owen thinks about the pumpkin's location. Let me follow the pumpkin's movements and who was present:......”
> > >
> > > Qwen3-32B-Reaosning “\<think\>\</think\>Answer: B”
> > >
> > > Qwen3-8B-Reasoning “\<think\>\</think\>Answer: B”
> > >
> > > We also report the performance of Qwen3-8B and Qwen3-32B using this method on HiToM.
> > > | Model | Non | Reasoning | Skip |
> > > | :--- | :--- | :--- | :--- |
> > > | **Qwen3-8B** | 0.558 | 0.481 | 0.506 |
> > > | **Qwen3-32B** | 0.586 | 0.68 | 0.529 |
> > >
> > > From the experiment, we can observe that the trend between reasoning v.s. non-reasoning is as the same as the reasoning v.s. skip. Our main conclusion about thinking still holds: reasoning does not always benefit ToM tasks. As this method is not universally applicable and will not extend our results beyond the current analysis (we have already included the comparison between Qwen3 reasoning and non-reasoning versions), we did not explicitly add it in our paper now. We would like to kindly ask the reviewer to provide the detailed steps or references that can help us implement it with other models. We are more than happy to include the results in our paper draft if we can apply this “prompt to skip thinking” method to all reasoning models like R1 and Claude-Sonnet-4.
> > >
> > > ## For W2+Q2
> > >
> > > We have included Sec. 4.3.1 as the supplementary point to the insight “Our analysis in Section 4.2.4 and 4.3.1 also supports this insight by examining the impact of constraining thinking tokens.” Our primary goal is to conduct an overall diagnosis and evaluation for reasoning models. In our paper, we identify the “slow thinking” and “option matching” as the two main patterns of reasoning failure, which are supported by the analysis in both Sec. 4.2 and Sec. 4.3. Besides, we introduced the error types in their reasoning strategies to further investigate the underlying mechanisms.
> > > We agree that more in-depth probing into the internal mechanism can be an important next step for understanding our diagnostic findings about reasoning models on ToM. However, it is beyond our scope for an overall diagnosis so we prefer to leave it for future work
> > >
> > > ## For citation
> > >
> > > Thanks for the suggestion. We have carefully checked all the citations and made sure they are correct.
> > >
> > > ## For experimental results
> > >
> > > | Model Name | Deployment Type |
> > > | :--- | :---: |
> > > | Claude-Sonnet-4 | API via Claude Official Platform |
> > > | Grok-3-mini | API via Azure|
> > > | DeepSeek-R1 | API via Azure|
> > > | DeepSeek-V3 | API via Azure|
> > > | GPT-4o | API via Azure|
> > > | GPT-o3 | API via Azure|
> > > | GPT-o4-mini | API via Azure|
> > > | Qwen3-8B | Local Deployment |
> > > | Qwen3-8B-Reasoning | Local Deployment|
> > > | Qwen3-32B | Local Deployment|
> > > | Qwen3-32B-Reasoning | Local Deployment|
> > > | Deepseek-R1-Distill-Qwen-7B | Local Deployment|
> > > | Deepseek-R1-Distill-Qwen-32B | Local Deployment|
> > > | Deepseek-R1-Distill-Llama-8B | Local Deployment|

---

### Author Response · Authors · 2025-12-03
**Summary of the rebuttal process**

We appreciate the efforts of all the reviewers, AC, SAC, and PC during the rebuttal session.

### To summarize our responses, we mainly clarify the following points:

**Control study:** Our main goal is to compare LRMs and LLMs to investigate the effectiveness of the reasoning ability. The comparison is two-fold. First, from a macro perspective, we can observe the overall performance difference between reasoning and non-reasoning models. Second, we also designed rigorous control studies to understand the findings from the micro level. The micro-level comparisons do not cover all models at the macro level since many of the reasoning models are closed models, making it impossible to compare with their non-reasoning versions with fully controlled studies.

**Contribution of the proposed method:** The main purpose of introducing this method is to further verify the problems we identified and to point out potential research avenues. Although these results are preliminary, they offer valuable support for our conclusions.

**Presentation on the insights:** We have carefully polished the presentation on our insights and reorganized them to be grounded in extensive experimental evidence. For example, for the insight "Slow Thinking Correlates with Reasoning Failures", we support it by the experimental investigation (Sec. 4.2.1, 4.2.2, 4.2.4) and method results (Sec. 4.3.1).

**Ethical statement:** We believe that fundamental understanding the mechanisms behind ToM is essential for building effective defenses. Our paper is a critical diagnosis rather than a capability advancement. We provide important insights on the ToM and reasoning ability. By comprehensively characterizing the patterns behind these failures, our work lays the groundwork for developing more transparent and reliable AI systems.

### In our revised paper, we have updated the following parts:

**Presentation:** We have comprehensively reorganized the experimental section and introduced "Takeaways" summaries to enhance readability and facilitate a better grasp of our core contributions. We also revised presentations for greater clarity. For example, we change the insight 1 to "Slow Thinking Correlates with Reasoning Failures." to more accurately reflect the underlying phenomenon.

**Insight:** We added a new insight "Moderate and Adaptive Reasoning Benefits Performance", which is supported by the evidence in Sec. 4.2.3, 4.2.4 and 4.3.1, to better synthesizes our experimental findings and suggests promising directions for the research community to optimize ToM reasoning.

**Experiments:** We added a new experiment in Sec. 4.2.4 to investigate the impact of different response lengths on the performance. It allows us to deeper analyze the relationship between performance and thinking process, support our insight "Moderate and Adaptive Reasoning Benefits Performance".

**Ethical statement:** We have revised the Ethical Statement to clarify our research position. We explicitly state that understanding the failure mechanisms of ToM is a prerequisite for safety, positioning our work as a critical diagnostic study that paves the way for more transparent and reliable AI systems.

During the rebuttal session, our score is **64442 => 66442**. We sincerely thank all the reviewers for the acknowledgment and constructive feedback.

---

### Meta-Review · Area_Chair_SSXf · 2025-12-17

**Summary:**

This paper investigates the relationship between theory of mind and reasoning models, and found that in general reasoning models do not consistently outperform non-reasoning models, and sometimes are weaker. This is a experimental paper with detailed experimental designs and analysis. However, the reviewers found a few significant methodological weaknesses. For example, Reviewer auky and Reviewer iDRt mentioned that comparing different model families introduces numerous confounding factors (e.g., training data, RL objectives) that prevent the isolation of reasoning effects. Furthermore, Reviewer VWm9 and Reviewer auky argued that the central claim connecting "slow thinking" to reasoning collapse establishes only correlation rather than causality, as task difficulty likely confounds both response length and accuracy. Finally, the proposed technical interventions were viewed by Reviewer VWm9 and Reviewer iDRt as preliminary and offering limited technical contribution. Therefore, I decide to reject this paper.

(I am ignoring the reviewer 3FPz, who raised the Ethics Concerns.)

**Reviewer Concerns:**

Addressed:
1. Reviewer Qmcm questioned why the Slow-to-Fast (S2F) experiments were limited only to Qwen models. The authors addressed this by adding experiments with R1-Distill-Llama-8B.
2. Reviewer auky requested clarification on whether models were accessed via API or locally. The authors successfully addressed this by providing a detailed table listing the deployment type (API vs. Local) for all 11 models.
3. Reviewer iDRt noted that most analysis focused heavily on the HiToM benchmark. The authors responded by including results from ToMATO and ToMBench.

Outstanding:
1. Reviewer auky remained unsatisfied with the comparison between different model families, arguing that confounding factors like training data and RL objectives were not isolated.
2. Reviewer VWm9 and Reviewer auky maintained that the relationship between long reasoning chains and failure is correlational, not causal.
3. Reviewer iDRt and Reviewer VWm9 mentioned that the proposed methods as being "straightforward" or "preliminary."

I am ignoring the reviewer 3FPz, who raised the Ethics Concerns.

**Reviewer Scores:**

I think Reviewer Qmcm raised the score from 4->6, but I do not think he/she is fully supportive this paper after reading the latest comment. For other reviewers, I believe they are not going to raise the scores, because there are many concerns about this paper that were resolved during rebuttal, and these concerns are not easy to resolve.

I am ignoring the reviewer 3FPz, who raised the Ethics Concerns.

---

### Decision · Program_Chairs · 2026-01-26

Reject